# Mechanistic Data Attribution:
# Tracing the Training Origins of Interpretable LLM Units

**Jianhui Chen** [* 1]   **Yuzhang Luo** [* 1]   **Liangming Pan** [1 2]

✉ jianhuichennlp@gmail.com   ⭘ Mechanistic-Data-Attribution

## Abstract

While mechanistic interpretability has identified interpretable circuits in large language models (LLMs), their causal origins in training data remain elusive. We introduce *mechanistic data attribution* (MDA), a scalable framework that employs influence functions to trace interpretable units back to specific training samples. Through extensive experiments on the Pythia family, we causally validate that targeted interventions—removing or augmenting a small fraction of high-influence samples—significantly modulates the emergence of interpretable heads, whereas random interventions show no effect. Our analysis reveals that repetitive structural data (e.g., LaTeX, XML) acts as a mechanistic catalyst. Furthermore, we observe that interventions targeting induction head formation induce a concurrent change in the model's in-context learning (ICL) capability. This provides direct causal evidence for the long-standing hypothesis regarding the functional link between induction heads and ICL. Finally, we propose a mechanistic data augmentation pipeline that consistently accelerates circuit convergence across model scales, providing a principled methodology for steering the developmental trajectories of LLMs.

## 1. Introduction

The rapid advancement and widespread deployment of large language models (LLMs) have transformed the landscape of artificial intelligence (Achiam et al., 2023; Yang et al., 2025;

---
*Equal contribution [1]State Key Laboratory of Multimedia Information Processing, School of Computer Science, Peking University [2]Beijing Academy of Artificial Intelligence, Beijing, China. Correspondence to: Liangming Pan <liangming-pan@pku.edu.cn>.

*Proceedings of the 43rd International Conference on Machine Learning*, Seoul, South Korea. PMLR 306, 2026. Copyright 2026 by the author(s).

**Stage 1 Localizing Interpretable Units**

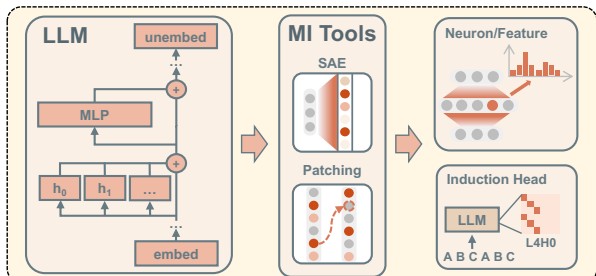

**Stage 2 Computing Data Influence with MDA**

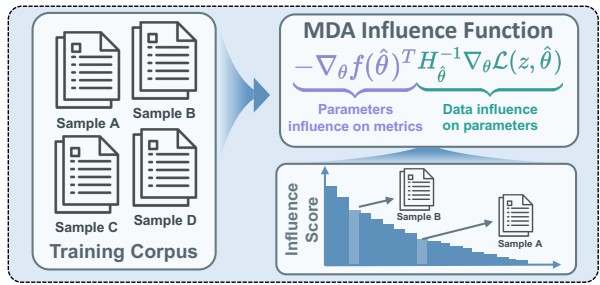

**Stage 3 Understanding & Intervening**

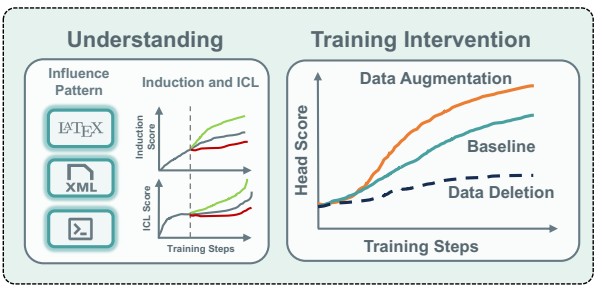

*Figure 1.* **The mechanistic data attribution (MDA) framework.** MDA identifies interpretable LLM units and quantifies the influence of individual training samples on their functional behavior. This enables both the discovery of mechanistic training dynamics and precise data-level interventions to steer model development.

Guo et al., 2025). This progress has been accompanied by a parallel surge in *mechanistic interpretability* (MI), a field dedicated to reverse-engineering these neural networks into human-understandable algorithms (Elhage et al., 2021). Recent efforts have successfully identified specific interpretable units within Transformer-based LLMs, such as in-

duction heads responsible for in-context copying and pattern completion (Olsson et al., 2022), knowledge neurons that store factual associations about specific entities (Dai et al., 2022), and mono-semantic features disentangled via sparse autoencoders (SAEs; Huben et al., 2023; Bricken et al., 2023). These findings effectively describe what a model computes during inference, offering a detailed anatomy of the model's internal mechanisms.

Despite these successes, current MI research remains predominantly *static*. While we can reverse-engineer what a circuit computes, we lack the tools to discern the *causal origins* of these computations within the training corpus. Bridging this gap holds significant value for both the scientific understanding of large language models and their practical governance. From a scientific perspective, identifying the data-driven origins of internal components provides a causal lens to observe how specialized circuits—such as those for logical reasoning (Hong et al., 2025) or factual recall (Nichani et al., 2025)—are shaped by the statistical properties of the training corpus (Chan et al., 2022). Simultaneously, from a practical standpoint, these insights enable precise data-level interventions. By filtering deleterious samples or augmenting high-leverage data, researchers can predictably modulate the emergence of specific mechanisms, offering more fine-grained control over internal representations compared to traditional data attribution methods (Koh & Liang, 2017; Grosse et al., 2023; Kou et al., 2025).

To address this challenge, we introduce *mechanistic data attribution* (MDA), a novel methodological framework designed to trace the training origins of internal mechanisms, as shown in Figure 1. Unlike traditional training data attribution (TDA) methods that typically focus on global model behavior (Koh & Liang, 2017; Grosse et al., 2023), MDA operates at the granularity of individual interpretable units, such as neurons, attention heads, or SAE features. By deriving a specialized formulation of influence functions (IF), our approach enables the precise computation of how specific training samples impact the functional properties of these units. This shift allows the analytical lens to move beyond descriptive analysis—asking "this circuit exists"—toward developmental tracing: explaining how "this data distribution caused the formation of this circuit." To be specific, we make the following four contributions in this paper:

• **Methodological Framework (Section 3):** We propose the MDA framework and derive a scalable, gradient-based approach leveraging IF. This framework enables the precise identification of training samples that most significantly impact the functional behavior of interpretable LLM units.

• **Causal Validation (Section 4):** We validate the causal effects of MDA through extensive data ablation and augmentation experiments during pre-training. Across four model scales in the Pythia family (Biderman et al., 2023) and two

distinct attention head types (induction and previous-token heads (Olsson et al., 2022)), we demonstrate that removing a small fraction ($\leq 10\%$) of high-influence samples from the total training set significantly hinders the emergence of targeted heads. Conversely, repeating these specific samples accelerates their formation, whereas randomly removing or augmenting the same number of other samples yields no such effect.

• **Mechanistic Insights (Section 5):** Our investigation into the formation of induction heads reveals several key findings: 1) *Data Composition*: Noisy data characterized by highly repetitive patterns, predominantly sourced from LaTeX and XML, significantly accelerates the emergence of induction heads. 2) *Transferability*: High-influence samples generalize effectively across different induction heads, exhibiting significant overlap in their identified influential subsets. 3) *Emergence Dynamics*: Induction head formation is not driven by sparse sample subsets but develops steadily as training tokens accumulate, with high-influence samples primarily modulating the *rate* of this process. 4) *ICL Correlation*: Enhancing induction head capabilities leads to a concurrent improvement in ICL performance, and vice versa. This bidirectional coupling provides causal evidence supporting the hypothesis that induction heads serve as a foundational mechanism for ICL (Olsson et al., 2022).

• **Practical Application (Section 6):** Leveraging these findings, we propose a practical data augmentation pipeline. This pipeline utilizes LLMs to extract patterns from high-influence samples and automatically generates code for data synthesis. Empirical results demonstrate that synthetic data generated via our smallest model generalizes effectively across various model sizes, consistently accelerating induction head formation. This provides a scalable methodology for fine-grained control of model behavior.

## 2. Related Work

### 2.1. Mechanistic Interpretability

Mechanistic interpretability aims to reverse-engineer neural networks into functional circuits that implement specific algorithmic behaviors (Elhage et al., 2021). Conventional research predominantly adopts a post-hoc paradigm, characterizing where and how circuits—such as induction heads (Olsson et al., 2022)—operate during inference. However, these analyses are largely static, treating internal mechanisms as fixed objects while overlooking their causal origins within the training corpus.

Recent research explores the developmental trajectory of model features, such as the emergence of induction heads (Tigges et al., 2024) and other mechanistic components (Ge et al., 2026). Complementing this, Chen et al. (2024) demonstrates that syntactic attention in masked language

model (MLM) emerges abruptly, using training-time interventions to modulate both structure and behavior. Others focus on controlled settings: Singh et al. (2024) and Kawata et al. (2025) use synthetic data and forward-pass interventions, while Aoyama et al. (2026) theoretically links induction head emergence to bigram statistics. While insightful, these approaches often rely on simplified data. In contrast, our work introduces a complementary mechanistic interpretability paradigm that directly traces the emergence of internal circuits back to specific training examples in realistic models trained on natural, unstructured data, providing a scalable framework for developmental tracing.

## 2.2. Training Data Attribution

Training data attribution aims to quantify the influence of specific training examples on model behavior, with foundational work establishing the use of IF in neural networks (Koh & Liang, 2017). To overcome the computational costs of Hessian-inverse calculations in LLMs, recent advancements leverage scalable approximations like EK-FAC (George et al., 2018). Building on these, Grosse et al. (2023) utilized IF to explain output likelihood through MLP layers, while other works focus on general behavioral abilities (Kou et al., 2025; Li & Sen, 2025). However, the influence of training data on intermediate functional components remains largely unexplored. A recent survey highlights the expanding scope of this field and underscores the necessity of bridging data attribution with mechanistic interpretability (Deng et al., 2025).

Beyond instance-level attribution, recent work has linked training data properties to model development. For instance, Qin et al. (2025) demonstrate that data complexity and diversity influence whether models adopt shortcut heuristics versus systematic rules. Similarly, studies have observed correlations between data patterns and the emergence of induction heads (Lee et al., 2026; Baker et al., 2026), yet these remain primarily observational. In contrast, we move beyond observation to conduct causal interventions, verifying the impact of specific data on circuit formation. This provides mechanistic insights into ICL and a framework for fine-grained training interventions.

# 3. Mechanistic Data Attribution Framework

We first introduce the essential preliminaries and then formally formulate the proposed MDA framework.

## 3.1. Preliminary

**Transformers and Induction Heads.** In Transformer-based language models, information flows via residual streams mediated by attention heads and MLP layers, many of which have been characterized as functionally interpretable units (Chen et al., 2025; Zhou et al., 2025). The most prominent among these are induction heads, which are considered critical components responsible for in-context learning capabilities (Olsson et al., 2022). Induction heads perform in-context copy-and-paste matching by tracking previous token transitions; specifically, they increase the prediction probability of $B$ whenever the sequential pattern $AB \ldots A$ appears in the previous context. Let $\theta$ denote the model parameters. We represent a specific component (e.g., an attention head $h$) by its corresponding subset of parameters $\theta_{\text{sub}} \subseteq \theta$.

**Influence Functions and EK-FAC Approximation.** IF provide a classic statistical tool to estimate the effect of upweighting a training sample $z_{\text{train}}$ on the loss of a test sample $z_{\text{test}}$. The influence score is given by:

$$\mathcal{I}(z_{\text{train}}, z_{\text{test}}) = -\nabla_\theta \mathcal{L}(z_{\text{train}})^\top H_\theta^{-1} \nabla_\theta \mathcal{L}(z_{\text{test}}) \quad (1)$$

where $H_\theta$ is the Hessian of the loss. Calculating the exact inverse Hessian is computationally prohibitive for LLMs. To scale this analysis, we employ the eigenvalue-corrected Kronecker-factored approximate curvature (EK-FAC) method (George et al., 2018; Grosse et al., 2023). EK-FAC approximates the Hessian layer-wise using Kronecker products of covariance matrices, enabling efficient estimation of the inverse-Hessian-vector product (IHVP) essential for attribution.

## 3.2. MDA Framework

While standard training data attribution (TDA) methods typically quantify the influence of training samples on the global model loss across the entire parameter space, we extend this paradigm by proposing the three-stage MDA framework (Figure 1). MDA enables the attribution of fine-grained, component-level behaviors to specific training data, allowing for a more localized and mechanistic understanding of model development.

**Stage 1: Localizing Interpretable Units.** The MDA framework is formally characterized by a three-tuple $(\mu, \pi, f_{\text{probe}})$. Specifically, we first define a monitoring metric $\mu$, which serves as a quantitative indicator for identifying specific interpretable units (e.g., the prefix-matching score for induction heads (Olsson et al., 2022)). Guided by this metric, we localize the target unit and isolate its associated parameter subspace $\theta_{\text{sub}}$ with the subspace projection $\pi$. Building upon a mechanistic understanding of the identified unit, we then design a probing function $f_{\text{probe}}$ (which may be identical to $\mu$) along with a corresponding evaluation dataset $\mathcal{D}_{\text{probe}}$ to assess the functional efficacy of the target unit. A summary of common design choices for $(\mu, \pi, f_{\text{probe}})$ across attention heads, neurons, and SAE features is presented in Table 3 (Appendix D).

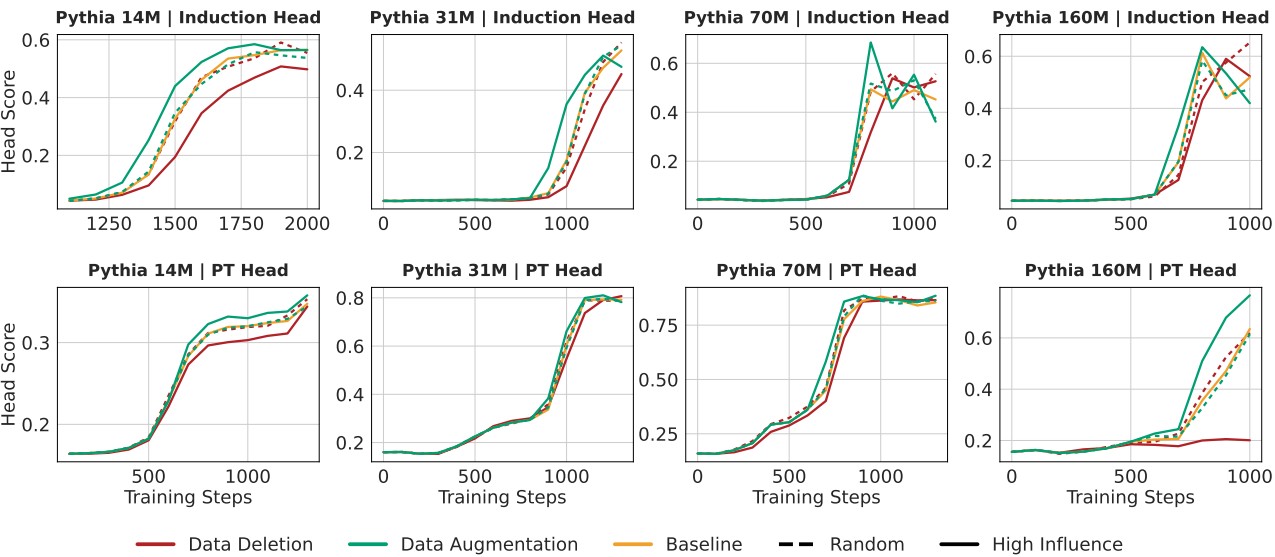

*Figure 2.* **Causal validation of MDA.** Intervened retraining shows that targeted deletion and augmentation of high-influence samples (identified via MDA) significantly modulate the emergence of induction and previous token (PT) heads. Head score is quantified via the prefix matching score metric from Olsson et al. (2022). Note the clear gap between MDA-guided interventions and the random baselines across different model scales.

**Stage 2: Unit-Specific Influence Calculation.** To capture the contribution of data to the behavior of the target unit rather than generic token prediction, we replace the standard validation loss $\mathcal{L}(z_{\text{test}})$ in Equation (1) with $f_{\text{probe}}(\theta, \mathcal{D}_{\text{probe}})$. Combining with the specified $\theta_{\text{sub}}$, the influence of a training sample $z$ on the interpretable unit is (formal derivation in Appendix A):

$$\mathcal{I}_{\text{MDA}}(z_{\text{train}}, \mathcal{D}_{\text{probe}}) = \\ - \nabla_{\theta_{\text{sub}}}\mathcal{L}(z_{\text{train}})^\top \hat{H}_{\theta_{\text{sub}}}^{-1} \nabla_{\theta_{\text{sub}}} f_{\text{probe}}(\theta, \mathcal{D}_{\text{probe}}) \quad (2)$$

where $\hat{H}_{\theta_{\text{sub}}}^{-1}$ is the EK-FAC-approximated inverse Hessian computed exclusively within the subspace $\theta_{\text{sub}}$. Algorithm 1 provides the full procedural details for the MDA calculation.

# 4. Causal Validation: Data Influence on Mechanistic Emergence

The effectiveness of MDA is validated through causal interventions on the Pythia model suite (Biderman et al., 2023), focusing on the emergence of induction heads and previous token heads (Olsson et al., 2022)—two foundational attention heads identified in LLMs. Specifically, the pre-training corpus is manipulated by either removing or duplicating high-influence samples discovered by MDA. This intervention allows us to directly assess the extent to which these specific training instances are causally responsible for the formation of the target units. To further demonstrate the generalizability of our findings beyond attention heads, we conduct preliminary experiments on SAE features, with

qualitative examples provided in Appendix L.3.

## 4.1. Experimental Setup

**Models and Target Units.** We conduct our experiments on the first four sizes of the Pythia suite (14M, 31M, 70M, 160M) and analyze two well-studied interpretable units: *induction heads* and *previous token heads*. Due to computational constraints, computing influence scores across the entire pre-training corpus is infeasible. Instead, we dedicate our attribution analysis to a critical developmental window $[t_{\text{start}}, t_{\text{end}}]$ that encompasses the emergence of these heads (detailed in Appendix C). Within this window, we use the full sequence of training data without sampling; this approach ensures we capture sparse yet pivotal training examples that may be essential for triggering mechanistic emergence, which stochastic sampling might otherwise omit.

**Causal Validation.** Although influence scores provide a theoretical proxy for data importance, they do not inherently imply causality. To rigorously establish the causal link, we perform bidirectional experiments via counterfactual retraining to evaluate both the *sufficiency* and *necessity* of the identified samples. Specifically, we conduct two intervention experiments: 1) Data Augmentation: high influence samples ($\leq 10\%$ of all samples) are duplicated and inserted at specific training steps; 2) Data Deletion: the gradients of high influence samples are masked during training. These experiments are localized within the $[t_{\text{start}}, t_{\text{end}}]$ window—either from scratch or continued from

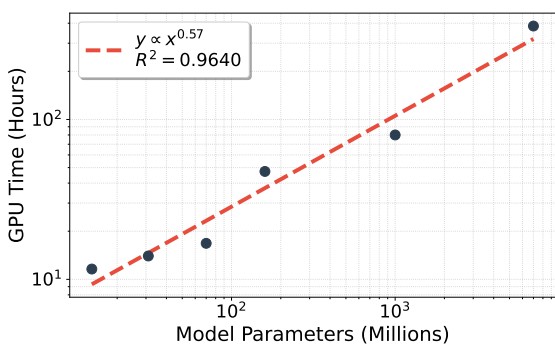

*Figure 3.* **Scalability of MDA.** MDA scales sub-linearly with model parameters. Both axes are on a log scale.

*Table 1.* **Representative high-influence training samples.** The top-ranked samples exhibit distinct repetitive structures across different domains.

| Type | Sample Content (Truncated) |
|---|---|
| **XML** | `<data>YnBsaX...</data>`
`<key>ANSIBright...</key>`
`<data>YnBsaX...` |
| **Code** | `public function`
`matches($subject): bool {`
`return $subject instanceof...` |
| **LaTeX** | `In the category of topological`
`spaces ($\mathbf{Top}$)...` |
| **Database** | `["44.2094250","28.6460842",`
`"Management","Bachelor"...` |
| **Meaningless** | `AczgAczgAczgAczgAczgAczg`
`AczgAczgAczgAczgAczgAczg` |

official checkpoints. To ensure reproducibility, all configurations, including hyperparameters, data sequencing, and random seeds, strictly adhere to the original Pythia repository. Detailed experimental configurations are provided in Appendix E.

### 4.2. MDA Identifies Causally Effective Data

As illustrated in Figure 2, data deletion consistently suppresses or delays head emergence, whereas random exclusion yields negligible impact—confirming these samples are necessary for circuit development. Conversely, data augmentation accelerates the phase transition relative to random insertion baselines, demonstrating their sufficiency in driving the targeted mechanism. Together, these results establish a robust causal link between MDA-identified training samples and the development of functional circuits. Additional validations and comprehensive robustness analyses—including alternative metrics, random seed stability, probe variants, and controlled baselines—are provided in Appendix G.

Furthermore, models under both augmentation and deletion regimes eventually converge to comparable saturation scores. This suggests that while specific samples modulate the emergence rate, the ultimate formation of these heads is a *collective property* of the general training distribution rather than being determined exclusively by a sparse subset. This aligns with Nanda et al. (2023), implying that induction circuits provide systematic loss reduction and receive consistent gradients across broad data. We further discuss this in Section 5.3. Notably, the *early drop* in induction scores during augmentation for Pythia-70M/160M is not an MDA failure, but a characteristic signal of accelerated emergence explored in Appendix I.

### 4.3. Generalizability and Scalability to Larger LLMs

While the controlled retraining experiments on the Pythia suite provide fine-grained mechanistic insights, a crucial question is whether MDA scales effectively to contempo-

rary, billion-parameter language models. To evaluate the generalizability and scalability of our framework beyond smaller architectures, we apply MDA tracing to the OLMo-2-1B and OLMo-2-7B models (Walsh et al., 2025).

Due to the immense computational cost of full retraining at this scale, we focus on the qualitative properties of the high-influence samples. The exact architectural and scaling configurations are documented in Appendix E. Notably, MDA exhibits sub-linear scalability in computational overhead as model parameters scale up (Figure 3). Furthermore, the top-ranked training samples (Figure 14) demonstrate that MDA successfully captures robust linguistic and semantic patterns even as the model scale increases by orders of magnitude.

## 5. Mechanistic Insights into Induction Head

In this section, we demonstrate how MDA serves as a complementary method to conventional post-hoc analysis, providing novel mechanistic insights into their formation and elucidating their functional coupling with the emergence of ICL capabilities.

### 5.1. Distributional Patterns of High Influence Data

We first examine the statistical distribution of induction head influence scores and the patterns of the top-ranked data. Our analysis focuses on samples with positive scores in the Pythia-14M model and the full distribution is provided in Appendix J.

**Power-Law Distribution of Influence.** Across all evaluated model scales, influence scores consistently exhibit a heavy-tailed distribution, as illustrated in Figure 4a. This distribution adheres to a distinct power-law, indicating that the emergence of mechanistic circuits is disproportionately driven by a sparse subset of high-leverage training signals.

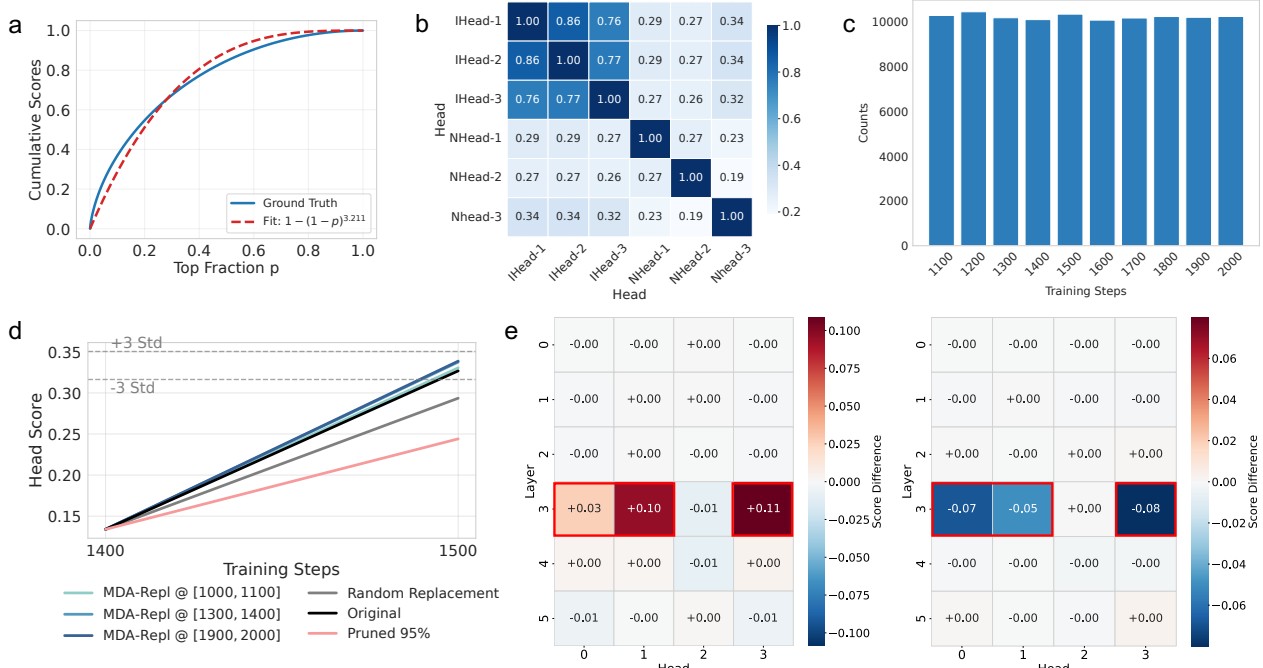

*Figure 4.* Distributional properties of high influence samples. **a) Power-law distribution:** The distribution of influence scores follows a power-law, where the top 10% of samples contribute up to 50% of the total cumulative influence. **b) Cross-head consistency:** High-influence samples identified by induction heads (`Ihead`) within the Pythia-14M model exhibit significant overlap, yet remain distinct from those identified by non-induction heads (`Nhead`). **c) Step uniformity:** The identified high-influence samples are distributed uniformly throughout the training corpus, showing no significant temporal clustering. **d) Induction head scores with high influence samples replaced with those from different steps.** MDA-Repl @ $[t_1, t_2]$ represents replaced by high influence sample in step $t_1$ to $t_2$. The random replacement baseline exhibits significant deviation from the MDA replacement, exceeding three standard errors ($3\sigma$). Pruned 95% means we randomly mask the gradient of 95% samples in training, while the induction head scores still show a non-trivial increase. **e) Induction scores differences of all heads from Pythia 14M.** High influence samples from one head are generalizable to other induction heads (red squares).

This concentration of influence provides an empirical justification for the intervention strategy employed in Section 4.

**Highly Repetitive Patterns.** Understanding the distributional properties that drive the emergence of specific mechanisms provides valuable empirical insights for optimizing model training (Chan et al., 2022). MDA offers a principled lens to identify functional patterns within unstructured training data. A qualitative inspection of samples with the highest positive influence scores (Table 1) reveals a previously under-explored finding: highly repetitive structures—including seemingly "noisy" or "garbage" sequences—act as primary catalysts for induction head formation. This observation aligns with the functional role of induction heads in predicting repetitive tokens within long-range contexts. For full examples, see Appendix L.

**5.2. Transferability of Influential Data**

To determine whether the identified training signals are unit-specific or mechanism-general, we analyze the overlap of high-influence samples across different functional components in Pythia-14M. Specifically, we compare the top three induction heads against three arbitrary non-induction heads. As illustrated by the block-diagonal pattern in Figure 4b, there is a pronounced overlap in influential data among different induction heads, indicating they are driven by a common set of mechanistic catalysts. In contrast, the overlap between induction and non-induction heads is notably lower. This dissociation confirms that MDA successfully isolates data specific to the *induction mechanism*, rather than merely identifying globally hard or high-loss samples. Furthermore, our analysis reveals that high-influence samples from a single induction head exhibit an impact on all induction heads that is an order of magnitude greater than on their non-induction counterparts (Figure 7). Crucially, this observational insight is reinforced by interventional evidence (via augmentation or deletion), where utilizing samples identified from an individual induction head strictly modulates the performance of other induction heads (Figure 4e). These results suggest that the identified data features are universally effective for the underlying mechanism itself, rather than being an artifact of a specific unit.

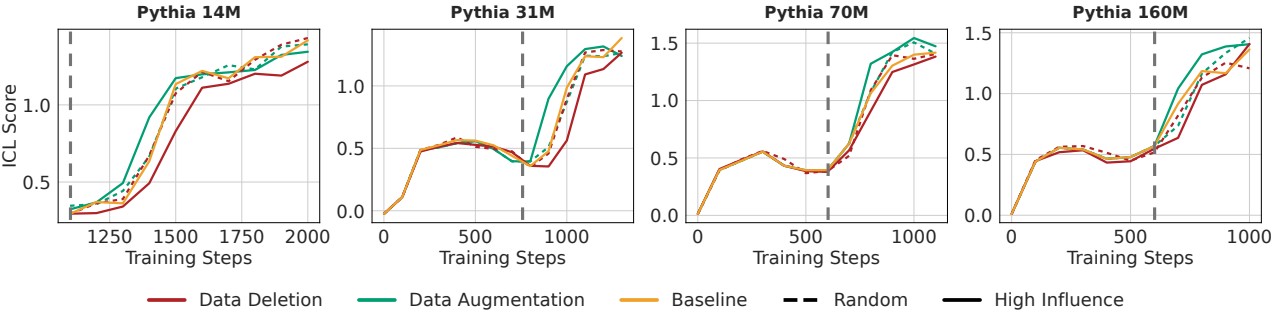

*Figure 5.* **Validating the functional role of induction heads in ICL via data intervention.** Under the same data augmentation and deletion settings used for induction heads (intervention steps denoted as grey dashes), the concurrent shifts in ICL scores (shown as absolute values for better visualization) and induction head strength provide causal evidence that these internal mechanisms are functionally coupled.

## 5.3. Emergence Dynamics

We first investigate the temporal dynamics of induction head formation by discretizing the training process into 100-step intervals. Surprisingly, we observe a remarkable *temporal homogeneity*: both the influential samples and their corresponding scores are distributed uniformly across the entire training trajectory (Figure 4c, Figure 13). This stands in stark contrast to the sharp *phase transition* observed in the induction scores (Figure 2), suggesting that the emergence of the mechanism is not tied to a sudden influx of specific data during the transition window.

To bridge this gap, we conducted a series of cross-stage controlled interventions. We replaced high-influence samples in the emergence window (steps 1400–1500) with those identified from early (1000–1100), mid (1300–1400), and late (1900–2000) stages. As shown in Figure 4d, high-influence samples from any training interval demonstrate universal effectiveness, consistently outperforming random baselines. Notably, the induction mechanism continues to develop even when 95% of the training samples within the window are masked.

These findings suggest that induction head formation follows a *steady accumulation* model rather than being triggered by a sparse subset of unique samples. Once training tokens reach a critical threshold, the phase transition occurs spontaneously. Within this framework, high-influence samples provide a higher signal density that shortens the required accumulation period.

## 5.4. Causal Link to In-Context Learning

It has been widely acknowledged that induction heads are usually correlated with ICL capabilities (Olsson et al., 2022). However, prior work has largely relied on observations. We provide direct causal evidence linking the development of induction heads to ICL capabilities with the help of MDA.

To rigorously quantify global ICL capability, we adopt the *ICL Score* metric proposed by Olsson et al. (2022), defined as the reduction in loss for late tokens compared to early tokens within a long context window: ICL Score = $\mathcal{L}_{500} - \text{mean}(\mathcal{L}_{0:50})$. A negative score indicates that the model is effectively utilizing the extended context to improve prediction accuracy relative to a shorter context. We evaluate this metric on WikiText-2 (Merity et al., 2017).

MDA enables precise intervention on induction heads, allowing us to rigorously verify their causal link to ICL. As shown in Figure 5, induction scores and ICL performance are tightly coupled: suppressing induction head formation degrades ICL, while enhancing them boosts proficiency. This aligns with Crosbie & Shutova (2025), who demonstrated that ablating induction heads impairs few-shot ICL; our results extend this by providing a training-time perspective, showing that modulating induction head formation directly governs ICL capability. While latent confounders remain a possibility, MDA offers a more controllable paradigm for mechanistic investigation than traditional observational studies.

## 6. Mechanistic Data Augmentation

Synthesizing the insights from Section 5—which established that induction heads are driven by specific structural motifs (e.g., frequent repetitions) rather than stochastic correlations—we propose *mechanistic data augmentation* as a practical approach for training data augmentation.

### 6.1. Data Augmentation Pipeline

We introduce a three-step pipeline that translates *post-hoc* attribution results into an *ante-hoc* training strategy. Our core insight is to automate the distillation of abstract structural patterns from the high-influence data identified by our framework and scale them up via procedural synthesis.

*Table 2.* Changes in induction head scores across various model sizes after augmenting the training set with Pythia-14M-guided synthetic patterns. [†] denotes augmenting with synthetic patterns from Pythia-160M.

| Model | Baseline (↑) | Augmentation (↑) | Δ |
|-------|--------------|------------------|------|
| 14M | 0.432 | 0.485 | +12.3% |
| 31M | 0.472 | 0.523 | +10.8% |
| 70M | 0.304 | 0.352 | +15.8% |
| 160M | 0.508 | 0.558 | +9.84% |
| 160M[†] | 0.508 | 0.521 | +2.56% |

**Step 1: Influence-Guided Sample Selection:** We employ the Pythia-14M model as a mechanistic proxy to identify high-leverage training data within the corpus. By executing the MDA framework during the 14M model's localized emergence window, we isolate the top-ranked $N = 2000$ training samples that exhibit the highest influence on circuit formation.

**Step 2: Automated Pattern Distillation via LLM:** To move beyond manual qualitative analysis, we leverage advanced LLMs (e.g., DeepSeek-V3 (DeepSeek-AI et al., 2025)) to automatically extract latent structural motifs from the identified data. We utilize a structured prompting strategy that tasks the LLM with analyzing batches of high-influence text and synthesizing them into rigorous JSON-formatted schemas.

**Step 3: Procedural Data Synthesis:** Based on the extracted JSON schemas, we prompt the LLM to generate *executable Python scripts*, which are subsequently used to programmatically synthesize training examples. This pipeline ensures that the synthetic data maintains strict structural consistency with the target mechanism while providing sufficient diversity in surface patterns. Crucially, this approach bypasses the need for computationally expensive large-scale corpus mining. Detailed prompts for all generation stages are provided in Appendix K.

### 6.2. Synthetic Data Generalize Across Model Scales

To assess the effectiveness and generalizability of our augmentation approach, we train four Pythia variants (14M, 31M, 70M, and 160M) by inserting mechanistic synthetic data during their localized emergence phases. This allows us to verify whether the synthetic data can consistently accelerate functional formation across different model scales.

We compare the induction head scores under synthetic augmentation against the baseline at the conclusion of the training interval. The results (Table 2) demonstrate that synthetic data consistently triggers and accelerates induction head formation across all model scales. This reinforces the hy-

pothesis that structural motifs, rather than specific semantic content, serve as the primary causal drivers of the induction mechanism. Remarkably, synthetic data identified from the 14M model exhibited efficacy on the 160M model that surpassed the data derived from the 160M model itself. This finding provides robust evidence for the cross-model consistency of mechanistic drivers, suggesting that the structural curriculum required to catalyze induction heads is scale-invariant. Such invariance validates the practical strategy of leveraging lightweight proxies to optimize the training of larger systems. The experimental details are provided in Appendix E.3.

### 6.3. Impact on General Performance

A potential failure mode of mechanistic-guided training data curation is the risk of distributional drift, where optimizing for specific circuit behaviors compromises broader task generalizability. To rule out this confounding factor, we evaluate the systemic side-effects of the MDA-based augmentation by contrasting it against standard pre-training baselines. Our evaluation protocols monitor two orthogonal axes of LLM capabilities: broad linguistic modeling over Wikitext-103 (Merity et al., 2017) and fine-grained open-domain factual retrieval via PopQA (Mallen et al., 2023).

As demonstrated in Figure 6a, the trajectories of MDA-augmented models closely overlap with those of the baselines across three Pythia models. Throughout the observation window, the augmentation pipeline causes no statistically discernible degradation in either language-modeling or factual retention capabilities. This finding suggests that the localized mechanistic enhancements do not compromise broader model quality.

### 6.4. Ablation Study on Insertion Strategy

To isolate the factors driving the success of our strategy, we performed an ablation study on the Pythia-14M model, focusing on two critical dimensions: insertion quantity ($N$) and insertion mode (concentrated vs. dispersed). By systematically varying these parameters, we characterize the optimal interventional regime required to maximize the acceleration of induction head formation. A comprehensive specification of these experimental settings is provided in Appendix H.

A comparison between synthetic and natural data performance (Figure 6b) reveals a fundamental trade-off between mechanistic density and semantic diversity: 1) *Small-Data Regime ($N \leq 50,000$):* Synthetic data outperform natural data. For instance, at $N = 12,500$ and $N = 25,000$, the synthetic insertion triggers a faster and sharper phase transition. This suggests that our synthetic templates possess a higher "causal density"—every sample is a perfect structural example, whereas high-influence realistic data may

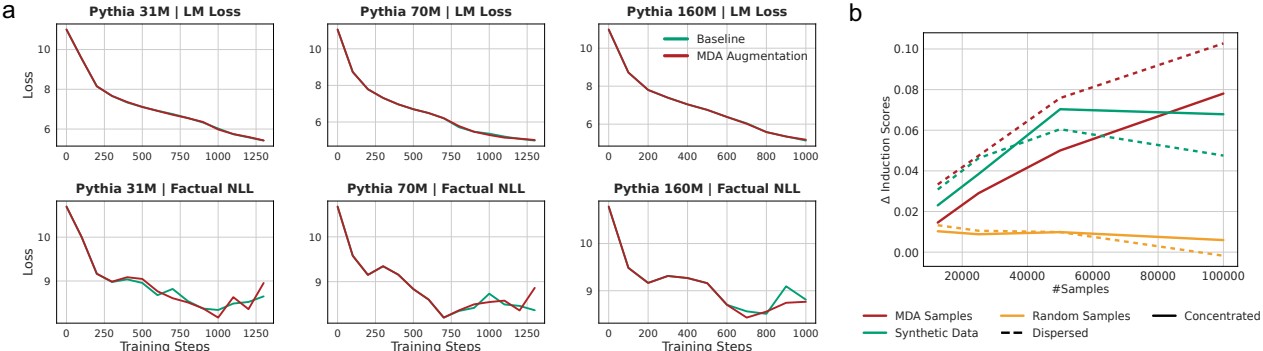

*Figure 6.* **a) Impact of MDA augmentation on general capabilities:** The closely aligned curves between the baseline and MDA augmentation across general language-modeling loss and factual-knowledge negative log-likelihood suggest that short-term MDA augmentation preserves long-term general model capabilities without degradation. **b) Different augmentation strategies on Pythia 14M:** For realistic data (red lines), augmentation effectiveness scales positively with sample size, with dispersed insertion consistently outperforming concentrated counterpart. For synthetic data (green lines), this trend holds true for smaller sample sizes but reverses in the large-sample regime.

still contain noise. 2) *Large-Data Regime ($N = 100,000$):* A crossover occurs where natural data begins to outperform synthetic data. We attribute this to *diversity exhaustion*. Since the synthetic data is generated from a finite set of extracted patterns, scaling to $100,000$ samples likely introduces diminishing returns due to excessive structural redundancy. In contrast, the natural high-influence samples, while noisier, offer a broader spectrum of lexical and syntactic variations, preventing overfitting to a rigid template and supporting sustained capability growth.

Regarding the insertion mode, dispersed insertion consistently outperforms concentrated insertion for natural data. By maintaining alignment with the original distribution, this approach minimizes optimization perturbations and avoids the convergence disruption often induced by concentrated gradient bursts. Spreading the data thus acts as a localized curriculum, facilitating stable mechanistic integration without disrupting concurrent feature acquisition. Interestingly, synthetic data exhibits divergent, scale-dependent results; we leave the investigation of this discrepancy to future work.

## 7. Discussions

**Scope of Attribution.** MDA relies on an underlying assumption that the attribution of an interpretable unit's behavior strictly accounts for the direct influence of the training data on that specific unit. In practice, however, training data can also exert indirect effects by modulating upstream nodes within the unit's computation graph. While an alternative formulation could aggregate the influence scores along the entire computational path, such an approach introduces substantial noise. For example, when a previous token head serves as an upstream component to an induction head, path-based aggregation fails to disentangle whether the computed

influence stems from the direct effect on the previous token head or the indirect effect propagated through the induction head. Functional decomposition methods like Chen et al. (2026b) may help determine the scope of attribution, and we leave the systematic decoupling of these cascading influences to future work.

**Mechanism vs Interpretable Units.** The term *mechanism* is frequently used interchangeably with the specific interpretable units identified as its basis. However, recent work has demonstrated that a single, abstract mechanism can be implemented by multiple structurally distinct circuits (Chen et al., 2026a). Supporting this view, our empirical findings reveal that high-influence samples for a specific induction head are universal across the broader induction mechanism (Figure 7). This phenomenon prompts a fundamental reconsideration of the relationship between abstract mechanisms and their concrete structural implementations, thereby leading to a deeper exploration of what constitutes a canonical causal mediator in mechanistic interpretability research.

## 8. Conclusion

We introduced MDA, a framework for tracing the causal origins of interpretable LLM mechanisms back to the training corpus. Our results demonstrate that the emergence of circuits, such as induction heads, is driven by identifiable data catalysts that generalize across model scales. By establishing a causal link between these internal mechanisms and macro-level capabilities like ICL, MDA provides a principled methodology for understanding LLMs. Furthermore, MDA provides a foundation for mechanistic alignment, enabling researchers to steer or unlearn specific model behaviors precisely.

## Impact Statement

This work presents the MDA framework that traces the functional origins of LLM circuits back to their training data. The potential societal impacts of this research are twofold. First, in terms of AI safety and governance, MDA provides a principled methodology for understanding how specific data distributions shape internal model behaviors. This enables more precise, data-level interventions to mitigate the emergence of biased or deleterious mechanisms, moving beyond superficial output-based filtering toward foundational transparency. Second, in terms of computational efficiency, our findings on data "catalysts" offer a path toward more efficient pre-training by identifying high-leverage data patterns, potentially reducing the carbon footprint of training large-scale models. While such attribution tools could theoretically be repurposed for targeted data poisoning, the transparency provided by MDA serves as a critical defensive layer, allowing researchers to audit and steer model development more responsibly.

## Acknowledgment

This work was supported in part by the Beijing Major Science and Technology Project under Contract No. Z251100008125054. This work was supported by the Beijing Academy of Artificial Intelligence (BAAI).

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

# A. Theoretical Background

### A.1. Derivation of Influence Functions

We adhere to the standard formalism of influence functions as introduced by Koh & Liang (2017). Let $z = (x, y)$ denote a training sample from the input space $\mathcal{X}$ and label space $\mathcal{Y}$. Let $\mathcal{L}(z, \theta)$ be the loss function for a model parameterized by $\theta \in \Theta \subseteq \mathbb{R}^p$.

Consider a training dataset $\mathcal{D} = \{z_1, \ldots, z_N\}$. The empirical risk minimizer $\hat{\theta}$ is given by:

$$\hat{\theta} = \arg\min_{\theta \in \Theta} \frac{1}{N} \sum_{i=1}^{N} \mathcal{L}(z_i, \theta). \tag{3}$$

To quantify the influence of a specific training example $z$ on the model parameters, we consider a perturbation where $z$ is upweighted by a small constant $\epsilon$. This corresponds to finding the minimizer of the perturbed empirical risk:

$$\hat{\theta}_{\epsilon,z} = \arg\min_{\theta \in \Theta} \left( \frac{1}{N} \sum_{i=1}^{N} \mathcal{L}(z_i, \theta) + \epsilon \mathcal{L}(z, \theta) \right). \tag{4}$$

The influence of the training point $z$ on the parameters is defined as the rate of change of the parameters with respect to $\epsilon$ at $\epsilon = 0$:

$$\mathcal{I}_{\text{params}}(z) \overset{\text{def}}{=} \left. \frac{d\hat{\theta}_{\epsilon,z}}{d\epsilon} \right|_{\epsilon=0}. \tag{5}$$

Since $\hat{\theta}_{\epsilon,z}$ is a minimizer, the gradient of the perturbed objective must be zero. Assuming the loss function is twice differentiable and strictly convex in the neighborhood of $\hat{\theta}$, the first-order optimality condition is:

$$\nabla_\theta \left( \frac{1}{N} \sum_{i=1}^{N} \mathcal{L}(z_i, \hat{\theta}_{\epsilon,z}) + \epsilon \mathcal{L}(z, \hat{\theta}_{\epsilon,z}) \right) = 0. \tag{6}$$

Let $R(\theta) = \frac{1}{N} \sum_{i=1}^{N} \mathcal{L}(z_i, \theta)$ denote the empirical risk. The condition simplifies to:

$$\nabla_\theta R(\hat{\theta}_{\epsilon,z}) + \epsilon \nabla_\theta \mathcal{L}(z, \hat{\theta}_{\epsilon,z}) = 0. \tag{7}$$

We perform a first-order Taylor expansion of the gradient $\nabla_\theta R(\hat{\theta}_{\epsilon,z})$ around the original optimum $\hat{\theta}$:

$$\nabla_\theta R(\hat{\theta}_{\epsilon,z}) \approx \nabla_\theta R(\hat{\theta}) + \nabla_\theta^2 R(\hat{\theta})(\hat{\theta}_{\epsilon,z} - \hat{\theta}). \tag{8}$$

Since $\hat{\theta}$ minimizes $R(\theta)$, we have $\nabla_\theta R(\hat{\theta}) = 0$. Let $H_{\hat{\theta}} = \nabla_\theta^2 R(\hat{\theta}) = \frac{1}{N} \sum_{i=1}^{N} \nabla_\theta^2 \mathcal{L}(z_i, \hat{\theta})$ denote the Hessian of the empirical risk. Substituting the expansion back into the optimality condition and keeping terms of order $O(\epsilon)$:

$$H_{\hat{\theta}}(\hat{\theta}_{\epsilon,z} - \hat{\theta}) + \epsilon \nabla_\theta \mathcal{L}(z, \hat{\theta}) \approx 0. \tag{9}$$

Solving for the parameter change $\Delta\theta = \hat{\theta}_{\epsilon,z} - \hat{\theta}$:

$$\hat{\theta}_{\epsilon,z} - \hat{\theta} \approx -\epsilon H_{\hat{\theta}}^{-1} \nabla_\theta \mathcal{L}(z, \hat{\theta}). \tag{10}$$

Dividing by $\epsilon$ and taking the limit $\epsilon \to 0$, we obtain the influence on parameters:

$$\mathcal{I}_{\text{params}}(z) = -H_{\hat{\theta}}^{-1} \nabla_\theta \mathcal{L}(z, \hat{\theta}). \tag{11}$$

Finally, to measure the influence of a training example $z$ on a specific target function $f(\theta)$ (e.g., the validation loss on a test point $z_{\text{test}}$, or in our case, the component-specific capability score), we apply the chain rule:

$$\mathcal{I}(z, f) = \frac{df(\hat{\theta}_{\epsilon,z})}{d\epsilon} = \nabla_\theta f(\hat{\theta})^\top \frac{d\hat{\theta}_{\epsilon,z}}{d\epsilon} = -\nabla_\theta f(\hat{\theta})^\top H_{\hat{\theta}}^{-1} \nabla_\theta \mathcal{L}(z, \hat{\theta}). \tag{12}$$

This formulation allows us to estimate how upweighting a training example $z$ affects any differentiable metric $f$ without retraining the model. In our methodology, $f$ represents the induction capability objective and the Hessian is restricted to the specific component subspace.

## A.2. Scalable Approximation via EK-FAC

Directly computing the Inverse-Hessian-Vector Product (IHVP) $H^{-1}v$ is computationally intractable for LLMs, as the Hessian matrix for a layer with dimensions $d_{in} \times d_{out}$ has size $(d_{in}d_{out})^2$. To address this, we employ the **Eigenvalue-corrected Kronecker-Factored Approximate Curvature (EK-FAC)** (George et al., 2018; Grosse et al., 2023) method , tailored to the specific parameter subspaces of induction and previous token heads.

**Standard K-FAC Assumption.** The K-FAC method approximates the Hessian of a linear layer (where $y = Wx$) by assuming independence between the input activations $x$ and the output gradients $g = \nabla_y \mathcal{L}$. Under this assumption, the Hessian block for weight $W$ decomposes into the Kronecker product of the input covariance $A$ and the gradient covariance $S$:

$$H \approx A \otimes S, \quad \text{where } A = \mathbb{E}[xx^\top] \text{ and } S = \mathbb{E}[gg^\top]. \tag{13}$$

This allows efficient inversion via the identity $(A \otimes S)^{-1} = A^{-1} \otimes S^{-1}$, reducing complexity from $O(d^6)$ to $O(d^3)$.

**Eigenvalue Correction (EK-FAC).** Standard K-FAC assumes that the eigenvectors of the Hessian are $U_A \otimes U_S$ and its eigenvalues are the Kronecker product of the eigenvalues of $A$ and $S$. However, this assumption is often inaccurate for neural networks, leading to poor curvature estimation. EK-FAC improves upon this by retaining the K-FAC eigenvector basis (which is generally a good approximation) but **correcting the eigenvalues**. Let $A = U_A \Sigma_A U_A^\top$ and $S = U_S \Sigma_S U_S^\top$ be the eigendecompositions of the covariance matrices. EK-FAC estimates the diagonal of the Hessian in this Kronecker basis:

$$H_{\text{EK-FAC}} = (U_A \otimes U_S)\Lambda(U_A \otimes U_S)^\top, \tag{14}$$

where $\Lambda$ is a diagonal matrix. The entries of $\Lambda$ are estimated efficiently via Monte Carlo sampling using the exact per-sample gradients projected onto the K-FAC basis. This correction captures the true scale of the curvature along the principal directions, significantly improving the accuracy of influence estimation.

**Joint Subspace Approximation for Attention Heads.** A critical modification in our methodology is the handling of the Query ($W_Q$) and Key ($W_K$) matrices, particularly for previous token heads. These matrices do not operate in isolation; the attention mechanism $Attention(Q, K, V) = \text{softmax}(\frac{QK^\top}{\sqrt{d_k}})V$ relies on the inner product of their outputs. Treating $W_Q$ and $W_K$ as independent blocks (i.e., a block-diagonal Hessian approximation) would enforce a zero interaction term $\frac{\partial^2 \mathcal{L}}{\partial W_Q \partial W_K} = 0$, ignoring the strong correlation between query and key updates.

To capture these essential correlations, we perform EK-FAC on the **concatenated joint subspace** $W_{joint} = [W_Q; W_K] \in \mathbb{R}^{(d_q+d_k) \times d_{model}}$.

- **Shared Input Covariance** ($A$)**:** Since both projections receive the same input $x$ (from the residual stream), the input covariance matrix $A \in \mathbb{R}^{d_{model} \times d_{model}}$ is computed once and shared.

- **Joint Gradient Covariance** ($S$)**:** The gradient covariance $S \in \mathbb{R}^{(d_q+d_k) \times (d_q+d_k)}$ is computed using the concatenated gradients $g_{joint} = [g_Q; g_K]$. Crucially, the off-diagonal blocks of this $S$ matrix capture the cross-covariance $\mathbb{E}[g_Q g_K^\top]$, effectively modeling the interaction between the query and key heads.

This fused approach ensures that the influence scores reflect the coupled nature of the attention pattern formation, rather than treating the query and key projections as disjoint feature extractors.

## A.3. Component-Specific Influence Formulations

Based on the derived influence framework and the EK-FAC approximation, we formally define the calculation of influence scores for the two specific types of mechanistic components investigated in this work.

**Case 1: Previous Token Heads (Attention Pattern Formation).** The primary function of a previous token head is to allocate attention mass to the immediately preceding token. This mechanism is governed solely by the interaction between the Query and Key projections, independent of the specific values being moved.

- **Target Objective $f_{\text{prev}}$ (Averaged over Probes):** To isolate the structural attention pattern from content-dependent interactions, we use a batch of $M$ random sequences $\mathcal{D}_{\text{probe}} = \{x^{(m)}\}_{m=1}^M$. We define the objective as the **average attention probability mass** assigned to the previous token position $(t-1)$ across all positions and all sequences in the batch:

$$f_{\text{prev}}(\theta) = \frac{1}{M} \sum_{m=1}^M \left( \frac{1}{T-1} \sum_{t=2}^T \mathcal{A}_{t,t-1}^{(\ell,h)}(x^{(m)}) \right). \tag{15}$$

Here, $\mathcal{A}_{t,t-1}^{(\ell,h)}(x^{(m)})$ denotes the attention score paid by the head to the token at $t-1$ for the $m$-th sequence. By averaging over random content, we ensure the gradient $\nabla f_{\text{prev}}$ encourages the formation of the specific *positional circuit* (i.e., attending to $-1$ offset) regardless of the token identities.

- **Parameter Subspace $\theta_{\text{prev}}$:** Since the attention pattern depends exclusively on $W_Q$ and $W_K$, we define the active parameter subspace as the concatenation of these two matrices:

$$\theta_{\text{prev}} = \text{vec}([W_Q; W_K]) \in \mathbb{R}^{2d_{model}d_k}. \tag{16}$$

- **Influence Score:** The influence of a training sample $z$ on the previous token head is computed as:

$$\mathcal{I}_{\text{prev}}(z) = -\nabla_{\theta_{\text{prev}}} f_{\text{prev}}(\theta)^\top H_{\text{EK-FAC}}^{-1}(\theta_{\text{prev}}) \nabla_{\theta_{\text{prev}}} \mathcal{L}(z, \theta). \tag{17}$$

Here, the gradient $\nabla_{\theta_{\text{prev}}} f_{\text{prev}}$ captures how the weights must change to sharpen the attention onto the previous token, while the Hessian accounts for the curvature of the joint Q-K manifold.

**Case 2: Induction Heads (Copy-Target Logit Contribution).** An induction head coordinates attention pattern formation (via Q, K) and content movement (via V, O). While the output is a differentiable function of all four matrices via the chain rule, we adopt a block-wise approximation for computational efficiency.

- **Target Objective $f_{\text{ind}}$ (Averaged over Copy-Target Probes):** To measure generalized induction behavior rather than memorization of specific tokens, we construct a set of synthetic repeated sequences $D_{\text{probe}} = \{x^{(m)}\}_{m=1}^M$. Each sequence is generated by sampling a random token block and repeating it across the context, producing many copy-target opportunities within a single sequence. For each query position $t$, we locate a previous occurrence of the current token,

$$\rho(t) = \max\{r < t : x_r = x_t\}.$$

If such a position exists and $\rho(t) + 1$ is valid, we define the copy target as

$$y_t = x_{\rho(t)+1}.$$

Let $\mathcal{V}(x)$ be the set of valid query positions. We define the induction probe as the direct logit contribution of the target head to these copy targets:

$$f_{\text{ind}}(\theta; D_{\text{probe}}) = \frac{1}{M} \sum_{m=1}^M \sum_{t \in \mathcal{V}(x^{(m)})} \left\langle r_t^{(\ell,h)}(x^{(m)}; \theta), W_U[:, y_t^{(m)}] \right\rangle,$$

where $r_t^{(\ell,h)}$ denotes the output of head $(\ell, h)$ at position $t$, projected back to the residual stream. This objective measures whether the head directly promotes the token following the previous occurrence of the current token, which is the standard copy-target behavior of induction heads.

- **Block-wise Decomposition:** This decomposition implies a block-diagonal assumption for the Hessian. We justify this independence because we have already explicitly captured the strongest parameter coupling—the multiplicative query-key interaction—within the joint $\theta_{QK}$ block, rendering the remaining second-order cross-correlations between the pattern and content pathways negligible for attribution purposes. Following this assumption, we decompose the parameter space into three orthogonal subspaces: $\theta_{QK} = \text{vec}([W_Q; W_K])$ for pattern formation, $\theta_V = \text{vec}(W_V)$ for value content, and $\theta_O = \text{vec}(W_O)$ for output projection.

- **Aggregated Influence Score:** The influence of a training sample $z$ is the sum of the influence scores computed independently within these subspaces:

$$\mathcal{I}_{\text{ind}}(z) = \mathcal{I}_{QK}(z) + \mathcal{I}_V(z) + \mathcal{I}_O(z). \tag{18}$$

Notably, we concatenate rather than multiply $W_Q$ and $W_K$ because the EK-FAC approximation is strictly derived for linear transformations of the form $y = Wx$. The concatenated projection $W_{joint} = [W_Q; W_K]$ preserves this linearity with respect to the input $x$, allowing for a valid Kronecker factorization of the curvature ($A \otimes S$). In contrast, formulating the influence in terms of the effective product matrix $W_{eff} = W_Q^\top W_K$ would render the attention scores quadratic with respect to the underlying parameters. This would violate the fundamental assumption of K-FAC, which relies on the gradient structure of linear layers, and would incorrectly model the optimization landscape of the actual trainable weights.

Regarding $W_V$ and $W_O$, although they theoretically form a composite linear map $\sum \mathcal{A}(W_O W_V x)$ due to the linearity of summation, we analyze them as distinct blocks for two reasons. First, from a statistical perspective, $W_V$ operates on raw token embeddings, while $W_O$ operates on *aggregated context vectors* post-attention. Fusing them would force the curvature approximation to rely solely on token-level covariance, ignoring the significant distributional shift and rank reduction caused by the attention mixing mechanism. Second, from a multi-head architecture perspective, $W_O$ acts as a global integration interface that projects the head's subspace back to the residual stream. By maintaining its separation, we allow the Hessian to capture the specific curvature of this re-projection step, which is distinct from the feature extraction role of $W_V$.

## B. From the perspective of Information geometry

### B.1. Proposition 1: Gradient Structure of Induction Heads

**Proposition 1.** *Consider a simplified single-head attention mechanism where the induction capability is measured by the attention probability assigned to a target induction token at index $j^*$ given a query at index $t$ (where $j^* = t - k$). The gradient of this objective with respect to the joint query-key parameter subspace $\theta_{QK}$ has a rank-1 outer-product structure between the query-side and key-side signals, weighted by the softmax residual $(\mathbf{1}[j = j^*] - \alpha_{tj})$.*

*Proof.* Let the pre-softmax attention score between query position $t$ and key position $j$ be denoted as $s_{tj}$. Under the joint parameterization $W_{QK} = [W_Q; W_K]$ and fixed input representations $x$, the score is given by the bilinear form:

$$s_{tj} = \frac{1}{\sqrt{d_k}} (W_Q x_t)^\top (W_K x_j). \tag{19}$$

The attention probability $\alpha_{tj}$ is obtained via the softmax function:

$$\alpha_{tj} = \frac{\exp(s_{tj})}{\sum_{i=1}^{t} \exp(s_{ti})}. \tag{20}$$

We define the induction objective $f_{\text{ind}}$ as the log-likelihood of attending to the correct previous token $j^*$:

$$f_{\text{ind}} = \log \alpha_{tj^*} = s_{tj^*} - \log \sum_{i=1}^{t} \exp(s_{ti}). \tag{21}$$

To find the influence direction, we compute the gradient with respect to the query parameter $W_Q$ (the derivation for $W_K$ is symmetric). Using the chain rule:

$$\nabla_{W_Q} f_{\text{ind}} = \sum_{j=1}^{t} \frac{\partial f_{\text{ind}}}{\partial s_{tj}} \frac{\partial s_{tj}}{\partial W_Q} \tag{22}$$

$$= \sum_{j=1}^{t} \left( \mathbf{1}_{\{j=j^*\}} - \alpha_{tj} \right) \frac{\partial s_{tj}}{\partial W_Q}. \tag{23}$$

Noting that

$$\nabla_{W_Q} s_{tj} = \frac{1}{\sqrt{d_k}} (W_K x_j) \, x_t^\top, \tag{24}$$

we substitute this back:

$$\nabla_{W_Q} f_{\text{ind}} = \frac{1}{\sqrt{d_k}} \left( (W_K x_{j^*}) - \sum_{j=1}^{t} \alpha_{tj}(W_K x_j) \right) x_t^\top \tag{25}$$

$$= \frac{1}{\sqrt{d_k}} W_K \left( x_{j^*} - \sum_{j=1}^{t} \alpha_{tj} x_j \right) x_t^\top \tag{26}$$

$$= \frac{1}{\sqrt{d_k}} W_K \left( x_{j^*} - \mathbb{E}_{j \sim \alpha_t}[x_j] \right) x_t^\top. \tag{27}$$

If we consider the joint $QK$ parameter block $\theta_{QK}$, the resulting gradient inherits a rank-1 outer-product form: a *query-side* factor (proportional to $x_t$ or $W_Q x_t$) multiplied by a *key-side residual* factor (proportional to $x_{j^*} - \mathbb{E}_{j \sim \alpha_t}[x_j]$ or its mapped version under $W_K$). In particular, when the softmax competition term is locally well-approximated as slowly varying (e.g., under small perturbations that do not substantially change the mass on competing keys), the ascent direction that *tends* to increase $\alpha_{tj^*}$ is aligned with directions that increase $s_{tj^*}$ relative to the other $s_{tj}$'s. Under such a local approximation, the component of the update that most directly increases the target score satisfies

$$\nabla_{W_{QK}} s_{tj^*} \ \propto \ \text{vec}(x_t \otimes x_{j^*}) \tag{28}$$

(up to the appropriate linear maps induced by $W_Q$ and $W_K$).

Thus, the influence score

$$\mathcal{I}(z) = -\nabla f_{\text{ind}}^\top H^{-1} \nabla \mathcal{L}(z) \tag{29}$$

can be interpreted as prioritizing training samples $z$ whose loss gradients $\nabla \mathcal{L}(z)$ have a large projection onto the (whitened) directions emphasized by $\nabla f_{\text{ind}}$ within the $QK$ block, i.e., directions spanned by rank-1 interactions between the query-side and key-side factors. Empirically, in many settings, such alignment is *often* associated with samples exhibiting token-to-token correspondences that resemble the probe's inductive pattern (e.g., local repetitions or copy-like structures).

## B.2. Proposition 2: Influence as Riemannian Projection

**Proposition 2.** *The component-specific influence score $\mathcal{I}(z)$ can be expressed as an inner product between the capability gradient and the sample gradient on the Riemannian manifold of statistical distributions, equipped with the Fisher Information metric.*

*Proof.* Let $\mathcal{P} = \{p_\theta : \theta \in \Theta\}$ be the manifold of probability distributions parameterized by the neural network weights $\theta$. The local geometry of this manifold is defined by the Fisher Information Matrix (FIM), $G(\theta)$, which serves as the Riemannian metric tensor:

$$G(\theta) = \mathbb{E}_{x \sim \mathcal{D}, \, y \sim p_\theta(\cdot|x)} \left[ \nabla_\theta \log p_\theta(y|x) \nabla_\theta \log p_\theta(y|x)^\top \right]. \tag{30}$$

Under the standard regularity conditions discussed previously (e.g., in well-specified models and/or near likelihood optima, where the curvature of the negative log-likelihood is well-approximated by Fisher-type metrics), we use the approximation $H \approx G$. Standard Euclidean gradient descent updates parameters as $\theta_{t+1} = \theta_t - \eta \nabla \mathcal{L}$. However, the steepest descent direction on the probability manifold, which minimizes the KL-divergence, is given by the **Natural Gradient** $\tilde{\nabla}$:

$$\tilde{\nabla} \mathcal{L} = G(\theta)^{-1} \nabla \mathcal{L}. \tag{31}$$

In our influence framework, we seek to quantify the impact of a sample $z$ on the target capability $f$. The first-order Taylor expansion of $f$ under a perturbation in the direction of the natural gradient of the loss $\mathcal{L}(z)$ is:

$$\delta f \approx \langle \nabla_\theta f, \delta\theta \rangle_{\text{Euclidean}} \tag{32}$$

$$= \langle \nabla_\theta f, \, -H^{-1} \nabla_\theta \mathcal{L}(z) \rangle \tag{33}$$

$$\approx -\nabla_\theta f^\top G(\theta)^{-1} \nabla_\theta \mathcal{L}(z). \tag{34}$$

We can rewrite this inner product using the Riemannian metric. Let $\langle u, v \rangle_G = u^\top G v$ denote the inner product on the tangent space $T_\theta \mathcal{P}$. The influence score becomes:

$$\mathcal{I}(z) \approx - \left\langle \nabla_\theta f, \ G^{-1} \nabla_\theta \mathcal{L}(z) \right\rangle \tag{35}$$

$$= - \left\langle G^{-1} \nabla_\theta f, \ G^{-1} \nabla_\theta \mathcal{L}(z) \right\rangle_G \tag{36}$$

$$= - \left\langle \tilde{\nabla} f, \ \tilde{\nabla} \mathcal{L}(z) \right\rangle_G. \tag{37}$$

**Conclusion:** The influence score is the negative inner product of the natural gradients of the mechanism probe $f$ and the training sample loss $\mathcal{L}$ on the statistical manifold. By using EK-FAC to approximate $G^{-1}$, we approximately whiten the parameter space, which can improve conditioning and reduce sensitivity to certain parameter scalings, thereby emphasizing the intrinsic alignment between the probe gradient and sample gradients within the chosen metric.

## C. Induction Attention Score and Formation Time

**Input construction.** Given a tokenized sequence $x_{1:L}$, we form a repeated input by concatenating $R$ identical copies of the same sequence:

$$x_{1:L}^{(1)} \parallel x_{1:L}^{(2)} \parallel \cdots \parallel x_{1:L}^{(R)}.$$

We evaluate the attention pattern of a target head $(\ell, h)$ on this repeated input and extract an induction-relevant stripe between consecutive repeats.

**Attention pattern.** Let $A_\theta^{(\ell,h)} \in [0,1]^{T \times T}$ denote the attention pattern (post-softmax attention weights) of head $(\ell, h)$ at parameters $\theta$, where $T = RL$ is the length of the repeated sequence. For a given repeat index $r \in \{2, \ldots, R\}$, we consider the query positions in the $r$-th block and key positions in the $(r-1)$-th block.

**Stripe extraction (diagonal with offset).** For each $r \in \{2, \ldots, R\}$, define the query range and key range

$$\mathcal{Q}_r = \{(r-1)L+1, \ldots, rL\}, \qquad \mathcal{K}_{r-1} = \{(r-2)L+1, \ldots, (r-1)L\}.$$

We extract the attention sub-matrix

$$B_r = A_\theta^{(\ell,h)}[\mathcal{Q}_r, \mathcal{K}_{r-1}] \in [0,1]^{L \times L},$$

and summarize induction behavior by the mean attention mass on its diagonal stripe with a fixed offset (e.g., offset $= 1$ for strictly repeated sequences):

$$s_r^{(\ell,h)}(\theta) = \frac{1}{|\mathcal{I}|} \sum_{i \in \mathcal{I}} (B_r)_{i,\, i+\Delta}, \qquad \Delta = 1,$$

where $\mathcal{I} = \{1, \ldots, L - \Delta\}$ indexes valid diagonal entries. Intuitively, this measures whether tokens in the current repeat attend to the corresponding shifted positions in the previous repeat, which is characteristic of induction-style copying.

**Dataset-level induction attention score.** We aggregate across repeats and across a dataset of sequences $\mathcal{D}$ to obtain a single scalar score per head and checkpoint:

$$s_{\text{ind}}^{(\ell,h)}(\theta) = \mathbb{E}_{x \sim \mathcal{D}} \left[ \frac{1}{R-1} \sum_{r=2}^{R} s_r^{(\ell,h)}(\theta) \right].$$

This score is directly computed from attention patterns and does not require additional supervision beyond the input sequences. Since we focus on early-stage emergence, we use a fixed repetition factor $R = 2$ throughout. Concretely, we first sample a base sequence length $L$ uniformly at random from the range $[8, 20]$ (in tokens), then draw a length-$L$ token sequence from the model vocabulary, and finally duplicate it once to form a length-$2L$ repeated input. We compute the diagonal-stripe attention score on the cross-repeat block of the resulting attention pattern. We report the dataset-level score by averaging over 100 independently constructed repeated sequences.

**Definition of Formation Window.** Let $\{\theta_t\}_{t=0}^T$ denote the sequence of training checkpoints. We define the *critical formation window* for an induction head $(\ell, h)$ as the interval $[t_{\text{start}}, t_{\text{end}}]$ encompassing its phase transition. Specifically, $t_{\text{start}}$ is identified as the checkpoint where the induction score $s_{\text{ind}}^{(\ell,h)}(\theta_t)$ diverges from the early-training noise floor (empirically $\approx 0.1$). Correspondingly, $t_{\text{end}}$ is defined as the point where the score reaches a functional sufficiency threshold. In our experiments, this window captures the sharp ascent of the induction score from its initial baseline to a stable level of 0.4–0.5, representing the distinct period during which the copy-paste mechanism is acquired.

Temporal Localization of Phase Transition Mechanistic components in LLMs often exhibit distinct developmental trajectories, characterized by a sudden **phase transition** rather than gradual improvement. Attributing data outside this critical developmental window introduces noise from unrelated model behaviors. Formally, for a target mechanism $\mathcal{M}$, we define a monitoring metric $\mu(t)$ that quantitatively reflects the mechanism's maturity at training step $t$. By tracking $\mu(t)$ throughout the training trajectory, we identify the critical interval $[t_{\text{start}}, t_{\text{end}}]$ where the mechanism emerges most rapidly. Our influence analysis is strictly constrained to the model checkpoints within this window.

## D. Detailed Framework Instantiation and Extensions

In this section, we provide the precise specifications used to instantiate the MDA framework for the mechanisms analyzed in the main text. We also discuss how the framework can be generalized to other interpretable units.

### D.1. Instantiation for Components and Mechanisms

Table 3 summarizes the mapping between the abstract framework components and the concrete properties of Induction Heads and Previous Token Heads. In addition to attention-head-based analyses, we also conduct preliminary experiments with sparse autoencoder (SAE) features. We include qualitative top-sample examples in Appendix L.3, which suggest that MDA can also be applied to feature-level mechanisms beyond individual attention heads.

### D.2. Mechanistic Data Attribution (MDA)

Algorithm 1 provides the full procedural details for the MDA calculation.

---

**Algorithm 1** Mechanistic Data Attribution (MDA)

---

1: **Input:** Pretrained Model parameters $\theta$, Target Component Subspace $\theta_{sub}$ (e.g., $W_{QK}$ of Head $L.H$), Synthetic Probe Data $\mathcal{D}_{syn}$, Training Dataset $\mathcal{D}_{train}$.
2: **Output:** Ranked Training Examples sorted by influence.
3: *// Phase 1: Unit-Specific Curvature Estimation*
4: Construct the EKFAC (via Equation (14))approximate inverse Hessian operator $\hat{H}_{\theta_{sub}}^{-1}$ estimated on a subset of $\mathcal{D}_{train}$.
5: *Note:* The curvature is strictly restricted to the parameters in $\theta_{sub}$ to filter out noise from other components.
6: *// Phase 2: Compute Mechanism Influence Vector*
7: Calculate the gradient of the mechanism-specific probe function on synthetic data:
8: $g_{probe} \leftarrow \nabla_{\theta_{sub}} f_{probe}(\mathcal{D}_{syn}, \theta)$
9: Compute the Inverse Hessian Vector Product (IHVP) effectively projecting the probe direction onto the data manifold:
10: $v_{IHVP} \leftarrow \hat{H}_{\theta_{sub}}^{-1} \cdot g_{probe}$
11: *// Phase 3: Score Training Data*
12: **for** each training sample $z_i \in \mathcal{D}_{train}$ **do**
13:     Compute gradient on training loss restricted to subspace:
14:     $g_{train}^{(i)} \leftarrow \nabla_{\theta_{sub}} \mathcal{L}_{train}(z_i, \theta)$
15:     Calculate influence score (projection):
16:     $s_i \leftarrow -(g_{train}^{(i)})^\top v_{IHVP}$
17: **end for**
18: **return** Top-K samples with highest scores $s_i$

---

*Table 3.* **Instantiation of the MDA Framework for Target Mechanisms.** We define distinct monitoring metrics $\mu$, probe functions $f_{probe}$, and parameter subspaces $\theta_{sub}$ tailored to the specific nature of different interpretable units.

| Mechanism | Monitoring Metric ($\mu$) | Probe Function ($f_{\textbf{probe}}$) | Subspace Projection ($\pi$) |
|---|---|---|---|
| **Induction Heads** | **Induction Score** (Prefix matching score on validation set). The critical window is defined where this score rises sharply. | **Copy-target direct logit contribution** on synthetic repeated sequences. *Rationale:* Measures the end-to-end information transmission capability. | $\mathbf{W_Q}, \mathbf{W_K}, \mathbf{W_V}, \mathbf{W_O}$ (Full Head) *Rationale:* Capture the coordination between attention pattern formation and value movement. |
| **Previous Token Heads** | **Previous-Token Score** (Average attention probability to token at $t-1$). | **Attention Score** allocated to the preceding token ($t-1$) on random sequences. *Rationale:* Isolates the specific attention pattern formation. | $\mathbf{W_Q}, \mathbf{W_K}$ (Query-Key Interaction) *Rationale:* The mechanism is primarily determined by local positional addressing. |
| **MLP Neurons** | **Activation Score** (Maximum or mean activation of neuron $i$ on trigger patterns). | **Neuron Activation** on specific trigger inputs. *Rationale:* Directly measures whether the neuron selectively responds to its hypothesized feature. | $W_{in}[:,i], W_{out}[i,:]$ (Input/Output Weights of Neuron $i$) *Rationale:* These parameters fully determine how the neuron reads from and writes to the residual stream. |
| **SAE Features** | **Feature Activation Score** (Average or peak activation of latent feature $k$ on trigger inputs). | **Reconstruction Fidelity** or latent activation magnitude for feature $k$. *Rationale:* Tests whether the latent feature reliably encodes the targeted concept. | $W_{enc}[:,k], W_{dec}[k,:]$ (Encoder/Decoder Weights of Feature $k$) *Rationale:* These weights define how the feature is extracted from and injected into model activations. |

# E. Detailed Experimental Setup

## E.1. Model Training and Checkpointing

To accurately capture the rapid phase transitions of mechanistic components, relying on standard open-source checkpoints (which are typically saved at coarse intervals, e.g., every 1000 or logarithmic steps) is insufficient. Therefore, we trained the first four sizes of the Pythia suite (14M, 31M, 70M, 160M) from scratch.

We strictly followed the official architecture and training hyperparameters provided by Biderman et al. (2023) to ensure our models are representative of the standard Pythia suite. The primary difference lies in our checkpointing strategy: we saved model states at a much higher frequency during the critical formation windows identified for each mechanism. All layer and head indices reported in this paper and the following tables follow a 0-based indexing convention.

## E.2. Configuration for Mechanistic Data Attribution

We provide the detailed hyperparameters used for the Mechanistic Data Attribution (MDA) framework in Table 4 (Induction Heads) and Table 5 (Previous Token Heads). The parameters include:

- **Target Component:** The specific Layer and Head index identified as the primary driver for the mechanism.

- **EKFAC Configuration:** The range of training steps $[t_{\text{start}}, t_{\text{end}}]$ and batch size used to estimate the covariance matrices ($\hat{H}^{-1}$).

- **Analysis Scope:** The total number of training samples (Num) scanned to compute influence scores.

- **Intervention Settings:** The specific training step where data augmentation was performed and the number of top-ranked samples (Top-K) selected for these interventions.

### E.3. Configuration for Mechanistic Data Augmentation

For the *Mechanistic Data Augmentation* experiments described in Section 6, we generated synthetic datasets based on the patterns extracted from the 14M model. To ensure a fair comparison, the position of synthetic data inserted was controlled. The specific insertion configurations are listed below:

- **14M:** Insert **100,000** synthetic samples at step **900**.

- **31M:** Insert **20,000** synthetic samples at step **800**.

- **70M:** Insert **20,000** synthetic samples at step **700**.

- **160M:** Insert **10,000** synthetic samples at step **600**.

Note that for larger models (31M-160M), we used a smaller volume of synthetic data compared to the natural data top-k insertion. This strict setting further validates the high causal density of the generated mechanistic patterns. As for the impact of the specific insertion quantity, we present a detailed investigation in Appendix H.

*Table 4.* **Experimental Configuration for Induction Heads.** The analysis covers the critical formation window specific to each model size. Influence scores were computed over a comprehensive set of samples (`Num`) without downsampling.

| Model | Layer | Head | Training Steps | EKFAC Batch Size | Analyzed Samples (`Num`) | Selected Top-K | Insertion Step |
|-------|-------|------|----------------|------------------|--------------------------|----------------|----------------|
| **14M** | 3 | 3 | 1000-1999 | 10 | 1,024,000 | 100,000 | 900 |
| **31M** | 4 | 3 | 0-1199 | 8 | 1,228,800 | 120,000 | 800 |
| **70M** | 4 | 3 | 0-999 | 6 | 1,024,000 | 100,000 | 700 |
| **160M** | 5 | 10 | 0-799 | 4 | 819,200 | 100,000[†] | 600 |

[†] For the 160M masking experiment, the exclusion count was set to 80,000.

*Table 5.* **Experimental Configuration for Previous Token Heads.** Similar to induction heads, specific layers and heads were targeted based on the Previous-Token Score.

| Model | Layer | Head | Training Steps | EKFAC Batch Size | Analyzed Samples (`Num`) | Selected Top-K | Insertion Step |
|-------|-------|------|----------------|------------------|--------------------------|----------------|----------------|
| **14M** | 2 | 2 | 0-1199 | 10 | 1,228,800 | 100,000 | 500 |
| **31M** | 3 | 0 | 0-1099 | 8 | 1,126,400 | 110,000 | 800 |
| **70M** | 3 | 3 | 0-899 | 6 | 921,600 | 80,000 | 600 |
| **160M** | 4 | 10 | 0-699 | 4 | 716,800 | 70,000[†] | 500 |

[†] For the 160M masking experiment, the exclusion count was set to 10,000.

### E.4. Hardware Configuration

All experiments were conducted on machines equipped with eight NVIDIA A100 GPUs. In total, approximately 800 GPU-hours were consumed for training and evaluation.

### E.5. Scaling

The main retraining-based validation experiments are conducted on models up to 160M parameters, primarily due to computational cost. These causal evaluations require repeated retraining and controlled comparisons, which are expensive at larger scales. However, MDA itself relies on scalable influence-function approximations, and its computation is restricted to mechanism-specific parameter subspaces rather than the full model. This allows us to edit the computation graph and substantially reduce tracing cost compared with full-model influence estimation. To further evaluate scalability, we additionally measure MDA tracing time on larger OLMo-2 models, including OLMo-2-1B and OLMo-2-7B. The runtime results in Table 6 show an approximately linear relationship between computational cost and model size. Since the tracing procedure is data-parallelizable, the wall-clock time can be further reduced with additional GPUs, making MDA practical for larger models on high-performance computing clusters.

*Table 6.* Runtime statistics for MDA tracing on different model sizes. The results suggest that the computational cost scales approximately linearly with model size, while the sample-wise tracing procedure remains fully parallelizable.

| Model | #Layer | #Head | Model Dim | GPU hours (A100) |
|---|---|---|---|---|
| **Pythia-14M** | 6 | 4 | 128 | 11.6 |
| **Pythia-31M** | 6 | 8 | 256 | 14 |
| **Pythia-70M** | 6 | 8 | 512 | 16.8 |
| **Pythia-160M** | 12 | 12 | 768 | 47.3 |
| **OLMo-2-1B** | 16 | 16 | 2048 | 80 |
| **OLMo-2-7B** | 32 | 32 | 4096 | 384 |

# F. Mechanism-Localized Attribution and Multi-Head Circuit Evolution

### F.1. Cross-Head Localization of Influence

Eq. (2) computes influence within the parameter subspace of a target unit, rather than over the full model parameter space. This makes MDA a direct, mechanism-localized approximation: it estimates how training samples affect the selected unit's mechanistic behavior, but does not attempt to decompose all indirect effects that may propagate through upstream or downstream components in the computation graph. We therefore empirically examine whether this localized attribution signal is concentrated on the intended mechanism, rather than being uniformly spread across unrelated heads.

To test this assumption, we perform a cross-head sanity check for the induction-head experiment. We first select the top 100,000 positive training samples according to the projection score of the target induction head L3H3. We then re-evaluate the same selected samples against all attention heads, using the corresponding head-specific projection scores. If the localized approximation were overly loose, these samples would receive comparable scores across many unrelated heads. In contrast, Figure 7 shows that the largest scores are concentrated on induction heads, while most non-induction heads receive substantially smaller scores.

This result supports a mechanism-localized interpretation of MDA. The selected samples are not merely globally high-loss or uniformly influential examples; instead, their influence is preferentially aligned with heads that implement induction-like behavior. Importantly, this does not imply that the selected samples influence only the single target head L3H3. Rather, it suggests that high-influence samples identified through one concrete unit can correspond to a broader abstract mechanism. This interpretation is consistent with the cross-head overlap and intervention transfer results in Section 5.2, where samples identified from one induction head generalize to other induction heads.

This distinction is important for interpreting MDA. Individual heads serve as tractable handles for localizing and probing mechanisms, but an abstract mechanism need not be implemented by a single unique unit. The same data catalysts may support multiple heads that instantiate similar functional behavior. Thus, Eq. (2) should be viewed as providing a scalable first-order approximation to the data origins of a target mechanism, rather than a complete causal mediation analysis over the entire computational graph. The multi-head dynamics discussed below further illustrate that, especially in larger models, the concrete implementation of a mechanism can involve interactions and redistribution across heads.

### F.2. Observations on Multi-Head Interaction and Circuit Evolution

In our primary analysis, we focused on the single strongest induction head to establish a clear causal link between training data and mechanism formation. However, induction circuits are rarely composed of isolated components; they often involve a distributed set of heads working in concert. In this section, we briefly discuss our preliminary observations regarding multi-head interactions and their evolution across model scales.

#### F.2.1. STABILITY IN SMALL MODELS (14M)

For the 14M model, we extended our analysis to monitor secondary induction heads (those with lower but significant induction scores) under the same intervention settings. We observed that (Figure 8) while the *magnitude* of strengthening or weakening varies among these heads—some are sensitive to data interventions while others are more resistant—their **functional identity remains stable**. Heads originally classified as induction heads consistently retain their "copy-paste" behavior throughout the training and intervention processes, and non-induction heads remain functionally distinct. This

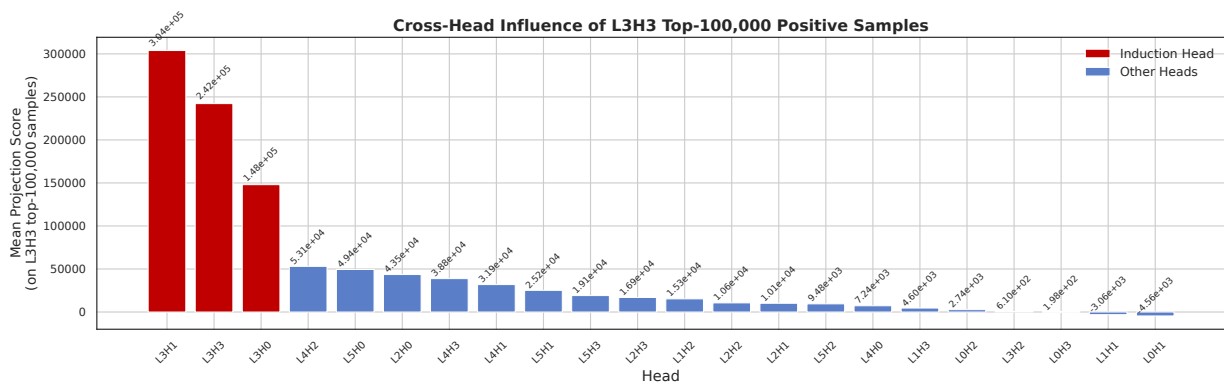

*Figure 7.* Cross-head projection scores on the top 100,000 positive samples selected by the L3H3 induction head. The projection signal is concentrated on induction heads, whereas most non-induction heads receive scores that are roughly an order of magnitude smaller. This separation supports the mechanism-localized interpretation of the approximation in Eq. (2).

suggests that in smaller architectures, the circuit topology is relatively static and rigid.

F.2.2. CIRCUIT RECONFIGURATION IN LARGER MODELS (160M)

In contrast, a divergent behavior emerges in larger models, such as Pythia-160M. Here, we observed instances where specific heads, initially identified as part of the induction circuit, completely ceased to exhibit(the one that shows a decrease of 0.26 in induction scores) induction behavior under certain interactive conditions or seemingly "handed off" their role to other components. This phenomenon implies that as model scale increases, the underlying circuit structure may undergo a form of **dynamic reconfiguration**. The functional role of a specific head is not as fixed as in the 14M model; instead, the circuit may evolve a more fluid topology where responsibilities are redistributed among a larger pool of redundant heads.

While we hypothesize that this reflects a sophisticated evolution in how larger models organize their internal mechanisms, a detailed characterization of these complex multi-head dynamics remains outside the scope of this work. We present this observation as an open question to stimulate future research into the scaling laws of circuit topology.

# G. Robustness and Additional Validation

We provide additional robustness analyses and validation experiments for MDA. These experiments address four questions: whether the ranking produced by MDA is stable across random seeds, whether the induction-head results can be explained by a simpler probe or by trivial n-gram repetition, whether MDA-based augmentation negatively affects broader model performance, and whether the observed induction attention patterns causally translate into correct next-token predictions.

## G.1. Ranking Stability across Random Seeds

We first evaluate the stability of the MDA ranking under different random seeds. Since MDA is used to identify the most influential training samples for a target mechanism, its practical usefulness depends on whether the top-ranked samples are robust to incidental randomness in the pipeline. As shown in Figure 9, the rankings obtained with different seeds remain highly consistent, indicating that MDA identifies a stable set of influential examples rather than seed-specific artifacts.

## G.2. Probe and Baseline Variants for Induction Heads

We further test whether the induction-head results depend on the specific choice of probe and whether they can be reduced to a trivial bigram-repetition effect. First, we replace our full-head induction probe with a QK-only prefix matching score. This variant only measures whether the head forms the appropriate prefix-matching attention pattern, without directly measuring whether the head writes the copied token through its OV pathways. As shown in Figure 10, this QK-only variant performs worse than the original MDA probe. This suggests that the $V$ and $O$ matrices play an important role in identifying training examples that actually promote the induction mechanism: matching the correct prefix alone is insufficient to capture the full copy-target behavior of an induction head. Second, we compare against a controlled data-augmentation baseline based

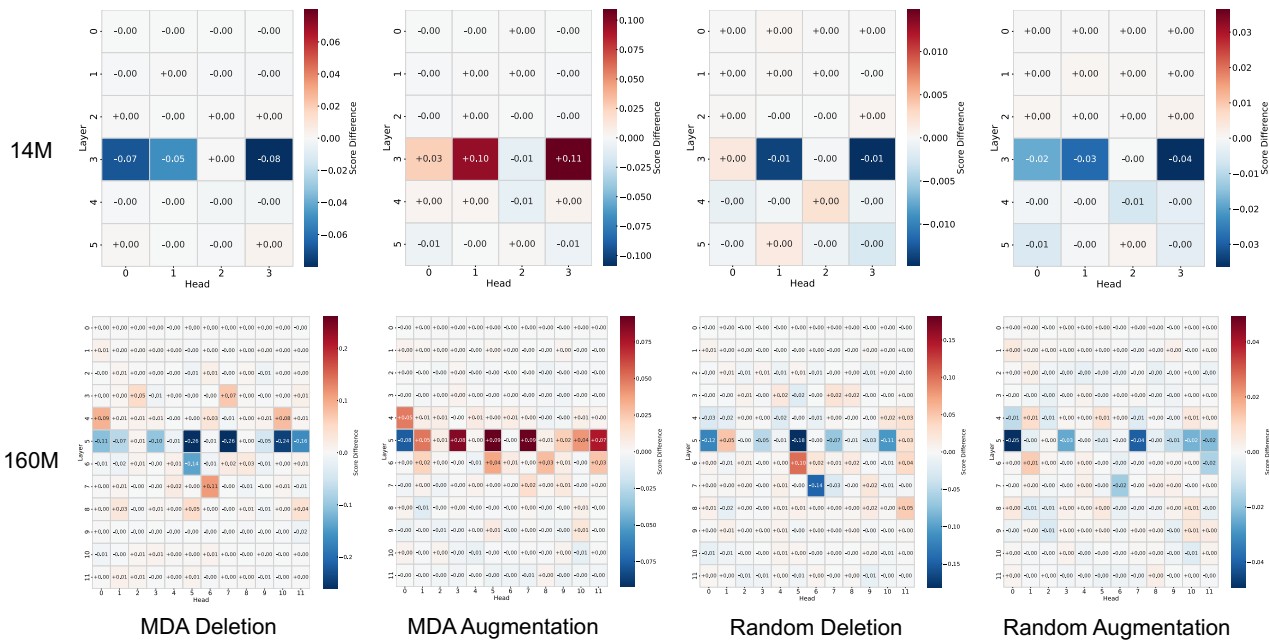

*Figure 8.* **The difference of induction scores among all heads for Pythia 14M and Pythia 160M.** For the 14M model, the maximum difference is limited to around 0.1, and the overall direction of change is largely consistent across heads. In contrast, the 160M model exhibits a form of compensatory behavior: while certain heads are significantly weakened, others are simultaneously strengthened. This suggests that, in larger models, interactions among heads are considerably more complex.

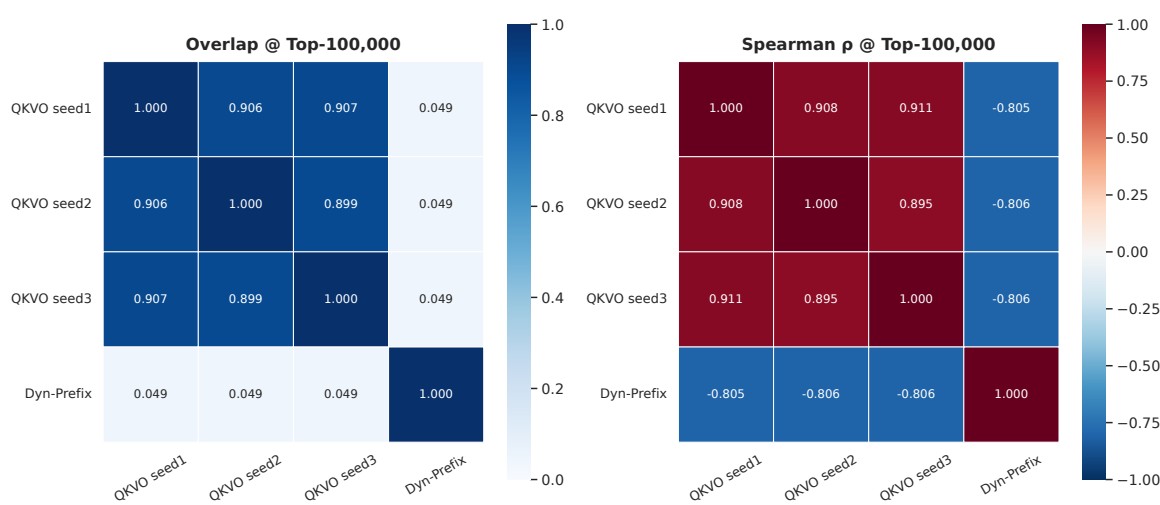

*Figure 9.* Overlap of MDA top rankings across different random seeds. Top samples remain highly consistent across seeds.

on n-gram repetition. Prior work has suggested that repeated bigrams can support the formation of induction heads, so a natural concern is that MDA may simply identify examples with repeated n-grams. To control for this possibility, we construct a baseline that uses the same inserted bigram and trigram as those found in the MDA-selected samples, but without using the MDA influence ranking itself. Other factors including entropy and domain are also included, forming a more rigorously controlled random baseline. However, the original MDA-selected augmentation still achieves the strongest effect.

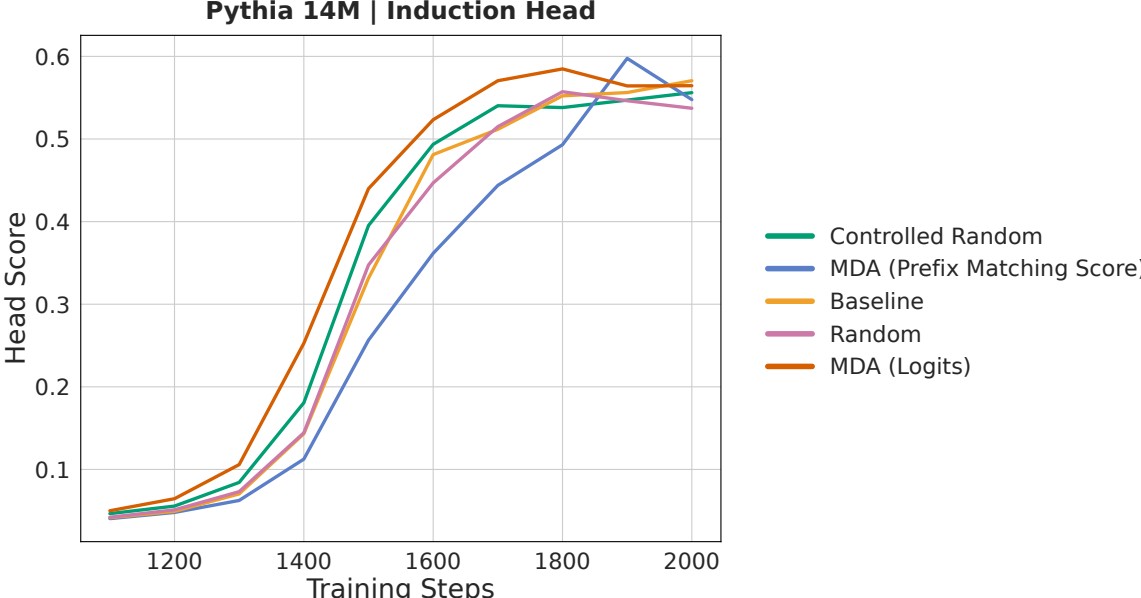

*Figure 10.* Robustness experiments for induction-head attribution. The original MDA(logits) pipeline achieves the strongest effect compared with a QK-only prefix matching probe(blue lines) and a controlled data augmentation baseline(Controlled Random).

This shows that MDA is not merely recovering a trivial repetition heuristic. Rather, it identifies training examples whose influence is specifically aligned with the target induction mechanism. Overall, these results support the robustness of our original MDA pipeline.

### G.3. Validation via Head-Specific Ablation Contribution

In the main text, we primarily monitored the *Prefix Matching Score* to track the emergence of induction heads. While this metric effectively captures the formation of the attention pattern, it is observational. To rigorously verify that these attention patterns causally translate into correct next-token predictions, we introduced a complementary interventional metric: Head-Specific Ablation Logit Contribution.

#### G.3.1. TASK DEFINITION AND METRIC CALCULATION

We evaluate the head's contribution on a synthetic "Induction Task" designed to strictly test the copy-paste capability. Specifically, we construct sequences with the structure:

$$[\texttt{Prefix}] \oplus [\texttt{A B C}] \oplus [\texttt{Gap}] \oplus [\texttt{A B}] \xrightarrow{\text{predict}} \texttt{C}$$

where [A B C] represents a unique random token pattern, and the model must rely on the context to predict C.

For a target head $h$ (e.g., Layer 3 Head 3 for the 14M model), we quantify its contribution as the drop in the correct token's logit when the head is functionally removed (zero-ablated). Let $\ell(C|x)$ be the logit of the correct token $C$ given input $x$. The ablation score $\Delta_h$ is defined as:

$$\Delta_h = \ell_{\text{clean}}(C|x) - \ell_{\text{ablated}}(C|x, h \leftarrow 0) \tag{38}$$

A positive $\Delta_h$ indicates that head $h$ positively contributes to the correct prediction.

#### G.3.2. CONSISTENCY OF RESULTS

We tracked this ablation score throughout the training process across our experimental configurations. As shown in Figure 11, the trajectory of the Ablation Logit Contribution exhibits a consistency with the observational Prefix Matching Score used in our main experiments. Both metrics capture the same phase transition interval and respond identically to our data

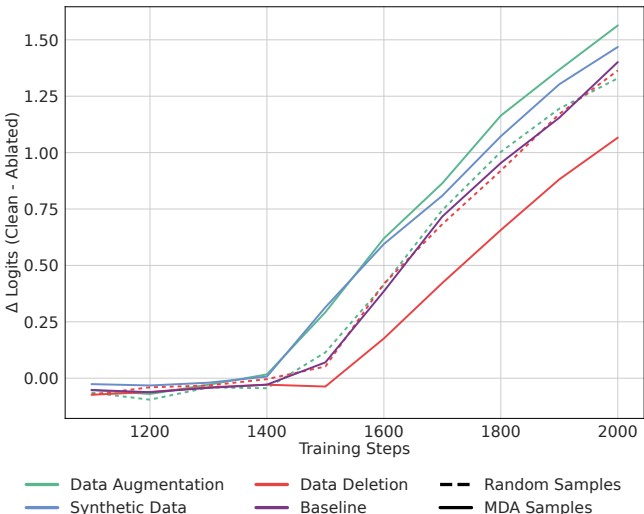

*Figure 11.* **Logit differences of Pythia-14M on the synthetic task.** Although logit difference and induction score are defined in fundamentally different ways, we observe a striking consistency between them: their ordering is fully aligned with the previously observed ordering of the prefix matching score. We interpret this mutual agreement as strong evidence that our method is reliable and robust.

interventions (Deletion and Augmentation). This alignment confirms that the "Induction Score" is a robust proxy: the identified heads are not merely attending to the correct history but are the causal drivers pushing the correct logits for in-context learning.

## H. Ablation Study on Insertion Dynamics

### H.1. Experimental Design

We designed a factorial experiment involving three data types, four quantity levels, and two scheduling strategies, resulting in a total of 25 experimental runs (including the baseline):

- **Data Types:**
    1. **Real High-Influence:** Top-ranked natural samples identified by MDA.
    2. **Synthetic Pattern-Based:** Data generated via the pipeline described in Section 6.1.
    3. **Random Control:** Randomly selected training samples.
- **Insertion Quantities ($N$):** We tested four volume levels: **12,500**, **25,000**, **50,000**, and **100,000** samples.
- **Insertion Schedules:**
    1. **Concentrated Injection (Burst):** All $N$ samples are inserted immediately after step 900.
    2. **Dispersed Injection (Uniform):** The $N$ samples are distributed uniformly across the interval from step 900 to 1400.

## I. Extended Training Dynamics and Window Selection

In our main causal verification results (Section 4), the induction score trajectories occasionally exhibit slight fluctuations or a minor decline after reaching their peak intensity. We devote this section to clarifying that this behavior is a natural characteristic of the model's optimization landscape rather than an artifact of our data interventions.

### I.1. Post-Peak Fluctuations

To provide context for the local behaviors observed in the critical window, we visualize the extended training trajectory of the 14M model, tracking the prefix matching score (Induction Score) from step 0 to step 3000 (Figure 12).

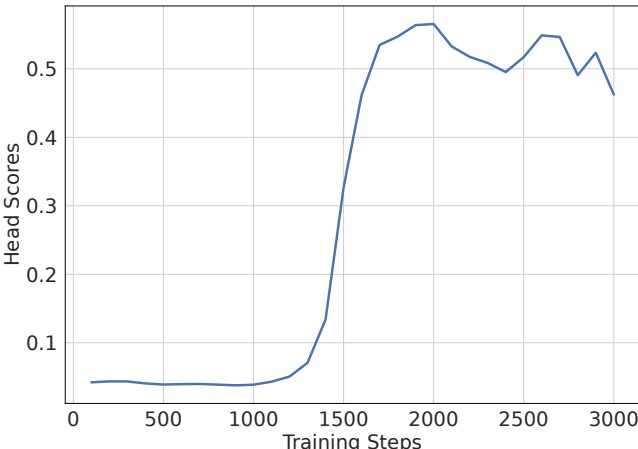

*Figure 12.* **Extended Training Dynamics of Pythia 14M.** From this figure, we can more clearly observe that the critical window we defined is highly pronounced. After step 2000, the score begins to exhibit small-scale fluctuations. We believe that the induction circuit has already been formed at this stage, and subsequent changes involve other trade-offs. Therefore, this regime is not further considered in the present study. This also accounts for the slight decline observed in the latter part of the curve in Figure 2.

As illustrated, the induction mechanism undergoes a dramatic phase transition between step 1200 and 2000. However, after this saturation point, the score does not remain perfectly monotonic. Instead, it naturally exhibits minor undulations and stabilization phases. This phenomenon likely arises from the complex interplay between competing optimization objectives: as the model begins to focus on minimizing loss for other linguistic features (e.g., complex syntax or factual knowledge), the parameters associated with the induction circuit may undergo slight adjustments, leading to the observed fluctuations. Therefore, the minor drops observed in our intervention experiments are consistent with the baseline dynamics.

### I.2. Justification for the Critical Window

Given the prohibitive computational cost of the Mechanistic Data Attribution (MDA) framework—which requires constructing high-dimensional curvature estimations and computing per-sample gradients—it is intractable to perform dense influence analysis over the entire training lifecycle.

Consequently, we strategically defined the Critical Window to encompass the period of maximum causal density: the phase transition where the mechanism originates. As verified by the extended trajectory, this window captures the most significant derivative of capability gain. Focusing our resources on this interval ensures that we identify the *formative* drivers of the mechanism, which is the primary research question of this work.

## J. Full Distribution of Influence Scores

In our main analysis, we focused primarily on the training examples with high *positive* influence scores, identifying them as the active drivers for the induction mechanism. In this section, to ensure a comprehensive understanding, we present the full spectrum of influence scores—including samples with negative influence (opponents)—using the 14M model as a representative case study.

### J.1. Net Positive Drive

We aggregated the influence scores for all training samples within the critical formation window. The global distribution of all samples via all suites is visualized in Figure 13. Crucially, we observe an asymmetry in magnitude: while there exists a subset of data that exhibits negative influence (theoretically hindering the mechanism), the cumulative sum of positive influence substantially exceeds the absolute sum of negative influence:

$$\sum_{z \in \mathcal{D}} \max(0, \mathcal{I}(z)) > \sum_{z \in \mathcal{D}} |\min(0, \mathcal{I}(z))|$$

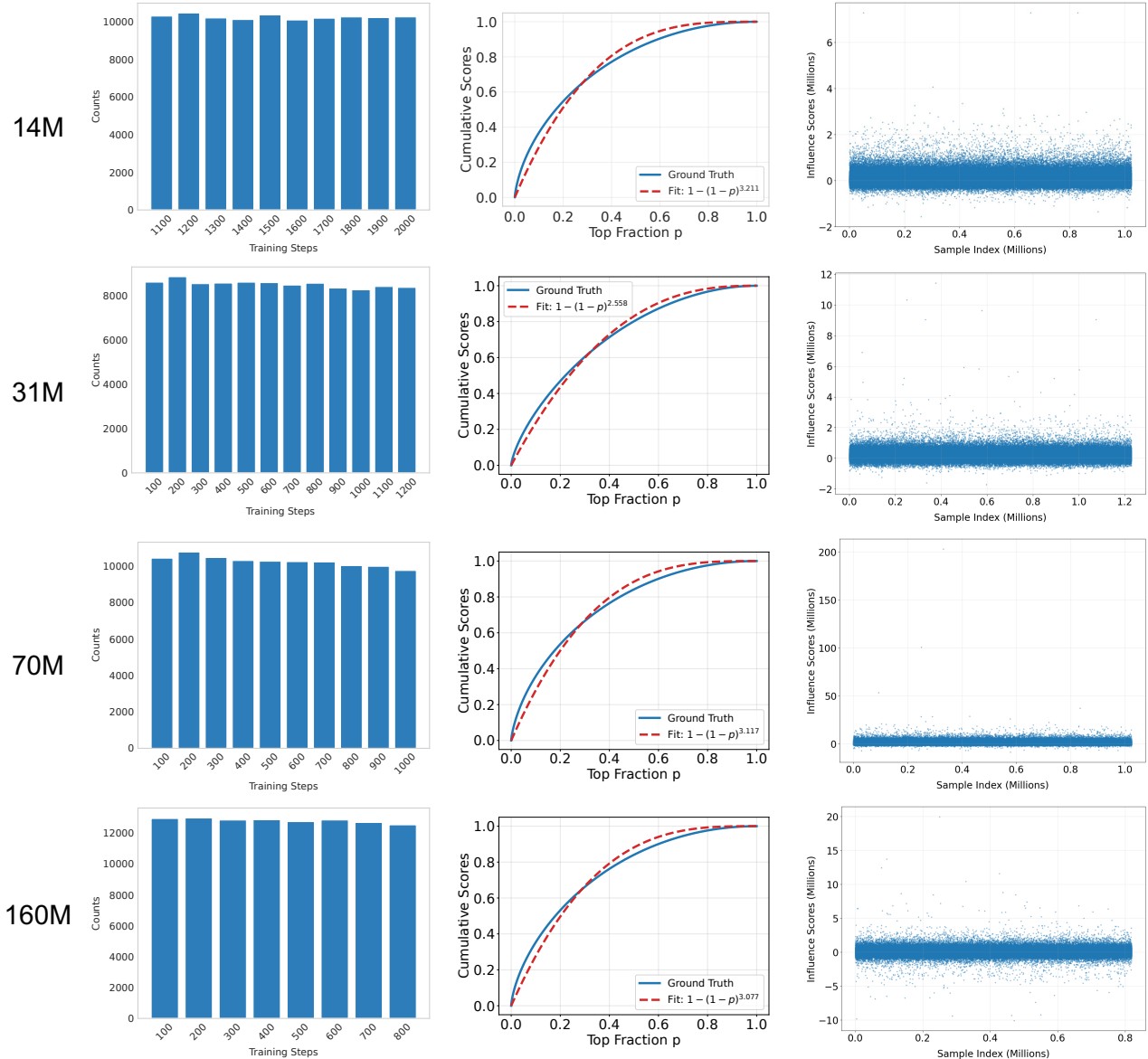

*Figure 13.* **The overall score distributions across all samples for all four models.** All four models exhibit a power-law behavior with an exponent close to 3. Moreover, the distributions are consistent across individual small bins, showing a uniform pattern.

This **net positive drive** provides the fundamental thermodynamic explanation for the circuit's emergence: despite conflicting gradients from various samples, the dataset on aggregate provides a dominant coherent signal favoring the formation of the induction heads.

### J.2. Robustness of Temporal Uniformity

Furthermore, we examined the temporal properties of this distribution. Consistent with the "uniformity" observation in Section 5.3, we find that this net positive ratio is maintained steadily throughout the critical window. Even when analyzing the data at varying temporal granularities (i.e., using different bin sizes or continuous sliding windows, rather than the specific intervals used in the main text), the distribution of influence mass remains remarkably uniform. This indicates that the force driving the phase transition is a **continuous, constant pressure** applied by the data distribution, rather than a transient shock caused by a specific anomaly batch. This goes along with the *lottery ticket hypothesis* for induction circuit proposed in Nanda et al. (2023).

# K. Implementation Details of Mechanistic Data Augmentation Pipeline

In this section, we detail the engineering pipeline used to operationalize the Mechanistic Data Augmentation strategy. The process is fully automated, converting raw high-influence training samples into executable data generation scripts via a three-stage workflow: (1) Sample Mining, (2) Pattern Extraction, and (3) Generator Implementation.

### K.1. Stage 1: Mining High-Influence Samples

We first extract the raw textual content of the samples identified by the MDA framework. As implemented in our data processing script, the procedure is as follows:

1. **Ranking:** We load the influence analysis results (Pickle format) from the 14M model and sort all training examples based on their *projection scores* in descending order.

2. **Filtering:** We select the Top-$K$ (where $K = 2000$) samples that exhibit the strongest positive influence.

3. **Decoding:** We decode the corresponding token indices back into human-readable text strings. These raw texts serve as the seed data for pattern extraction.

### K.2. Stage 2: LLM-Driven Pattern Extraction

To convert unstructured raw text into structured rules, we employ **DeepSeek-V3** as a pattern extraction engine. We feed the mined text samples into the LLM with a specialized system prompt:

> **System Instruction:**
> You are an expert in linguistic pattern recognition. Analyze the provided text samples and extract their underlying structural templates. Ignore specific semantic content and focus on the fixed "mechanistic" structure.
> **Output Requirement:**
> Return a valid JSON object with the following schema:

```
{
    "pattern_id": "Unique identifier (e.g., 'p001')",
    "pattern_name": "Short descriptive name",
    "anchor_tokens": ["List of invariant strings, e.g., 'Chapter', 'Step'"],
    "fields": [
        {
            "name": "Variable placeholder name used in template",
            "type": "Type of content (e.g., 'fixed_list', 'random_text')",
            "values_or_rules": ["List of options"] or "Description of generation rule"
        }
    ],
    "template": "Global format string with placeholders (e.g., '{anchor}')",
    "length_control": "Constraints to match the original token length"
}
```

This step produces a merged JSON registry containing definitions for all identified mechanistic templates. For 14M, it successfully extracted around 900 unique (deduplicated) patterns.

### K.3. Stage 3: Automated Generator Implementation

In the final stage, we automatically convert the static JSON schemas into executable Python generation functions. This is achieved through a meta-programming script that orchestrates the following steps:

**Meta-Prompting for Code Generation.** We iterate through each pattern in the JSON registry and construct a prompt for DeepSeek-V3. The prompt explicitly requires the model to write a robust Python function that satisfies the schema constraints. The core prompt template used is:

> **System:** You are a Python code generation expert.
> **User:** Please write a Python generation function for the following data pattern.
> **Requirements:**

1. Function Name: `generate_<pattern_id>_<name>`
2. **Fields Rule:** Strictly generate data based on the `values_or_rules` and `type` defined in the `fields` list.
3. **Template Structure:** The output string must strictly follow the defined `template`.
4. **Length Control:** Implement looping logic to ensure the output length approximates `target_tokens`.

**Pattern Definition JSON:** `<Insert JSON Content>`

# L. Qualitative Inspection of High-Influence Samples

To provide concrete intuition regarding the data drivers identified by the MDA framework, we present a qualitative inspection of the top-ranked training samples. We first inspect attention-head samples for Pythia 14M and larger OLMo models, and then provide preliminary SAE-based examples as a qualitative extension beyond attention heads. As discussed in Section 5.1, our analysis indicates that induction heads are primarily driven by data containing long-range repetitive structures.

### L.1. Attention-Head Samples in Pythia 14M

Table 7 and Table 8 displays representative high-influence examples explicitly mined from the 14M model's training corpus. These samples, selected from the top 0.1% of the influence distribution, span diverse modalities including structured code (XML/Base64), raw binary-like sequences, and domain enumeration lists. Despite their superficial differences in format, they all share a robust *mechanistic signature*: a specific pattern or token sequence (highlighted in bold) appears in the context and is repeated after a variable interval, providing a strong "copy-paste" supervision signal for the induction circuit.

### L.2. Scaling Attention-Head Attribution in OLMo

We also inspect top-ranked samples for induction heads in larger OLMo models. Figure 14 shows qualitative examples from OLMo-2-1B and OLMo-2-7B. It is worth noting that on larger models, compared to what Pythia-14M demonstrated previously, the top samples often appear more abstract and structured: the samples include repeated phrase templates, tabular month-value sequences, query-template lists, code-like symbol patterns, and HTML/CSS fragments. This qualitative shift suggests that as model scale increases, the notion of "repetition" captured by induction-related heads may become less tied to exact token-level repetition and more aligned with higher-level structural recurrence.

### L.3. Preliminary SAE-Based Top Samples

While our main experiments focus on attention-head mechanisms, MDA is not restricted to head-level probes. As a preliminary extension, we apply the same attribution perspective to sparse autoencoder (SAE) features and inspect the top-ranked training samples associated with selected features. These examples provide a qualitative sanity check that MDA can identify data patterns aligned with interpretable feature-level behavior. Figure 15 shows examples from Pythia-70M SAE features. The top samples for Feature 9500 contain LaTeX-like mathematical expressions and symbolic notation, while Feature 31939 is associated with newline- or formatting-heavy text. These examples suggest that SAE features can capture recurring textual formats or syntactic templates, and that MDA can retrieve training samples that instantiate these patterns.

| **OLMo-2-1B** |
| --- |
| assies embassy embattled embedded embedding embezzlement emblematic embodied embodiment embody emboldened embolism |
| @kpop14young @GUMMYB34RZz @JohnEvans @staceyholley @MaricelvaRomero @EmilySavage @EmilyGardner @HuonTreeRoo @CreeTheOtaku |
| \nbody image, body type\nBody type, fitness, exercise, sports\nbodyweight exercises\nbones, food for healhy bones, calcum, strong bones |
| subtitles subtle subtract subtracted suburb suburban suburbs subversive subway subways succeed succeeded |
| Taken 2002 ep02\nTaken 2002 ep03\nTaken 2002 ep04\nTaken 2002 ep05\nTaken 2002 ep06\nTaken 2002 ep07\nTaken 2002 ep08 |
| **OLMo-2-7B** |
| chewable tablets, magnesium bydroxide chewable tablets, magnesium nydroxide chewable tablets, magnesium jydroxide chewable tablets |
| 14202\nDec-10 25103\nNov-10 28146\nOct-10 44141\nSep-10 46128\nAug-10 90273\nJul-10 37497\nJun-10 49127\nMay-10 48231 |
| thing stock,\nfashion stocks,\npublicly traded fashion companies,\nclothing company stocks,\napparel stock |
| a03, a24, \\\n          a40, a11, a32, a02, a23, a44, a10, a31, a01, a22, a43, a14, a30\n#define P6 a00, a02, a04 |
| 12, 13, 0.33) !important;<!– [et_pb_line_break_holder] –>}<!– [et_pb_line_break_holder] –>/* change order |

*Figure 14.* Top-ranked samples for induction heads in OLMo-2-1B and OLMo-2-7B. Compared with Pythia 14M, the OLMo samples suggest a more abstract form of repetition, including repeated templates, tabular structures, code-like patterns, and markup fragments.

| **Pythia-70M SAE** |
| --- |
| **LaTeX (Feature 9500)** |
| i_{2}}))}_{t}=f^{(u_{i_{k}})}_{t+s}\n=a_{t+u_{i_{2}}-1}$, $1\\leq t<i_{2}-u_{i_{2}}+1$ |
| \\oplus\\liea'$ where $\\liea'\\subset \\lieq'$. Again we have the fact that $\\exp(\\lieh)\\cdot |
| BG$, and for every $c:\\bG_m\\to \\BC^*$ the object $$\\BW_{X,\\bx,c}:=(\\wh\\BW_{X,\\bx}\\otimes \\BC) |
| **\n (Feature 31939)** |
| Calculate -6 divided by 7.**\n**-6/7**\n**Calculate 0 divided by 234.**\n**0**\n** |
| the table.**\n\n**#**\n\n**# The Kryptea Do Not Knock**\n\n**\"What happened to Vinjar?\" |
| See also_ bipedalism; hand, human**\n\n**_Australopithecus anamensis. See also_ bipedalism**\n\n**Australopithecus garhi. |

*Figure 15.* Top-ranked samples for selected Pythia-70M SAE features. Feature 9500 is associated with LaTeX-like mathematical expressions and symbolic notation, while Feature 31939 is associated with newline- or formatting-heavy text. These qualitative examples suggest that SAE-level MDA can retrieve training samples aligned with interpretable feature-level patterns.

*Table 7.* **Representative High-Influence Training Samples.** We showcase three actual top-ranked examples mined from the training corpus. They exhibit distinct long-range repetitive structures (highlighted in bold) across different modalities.

| Type | Sample Content (Truncated) |
| --- | --- |
| XML | <div>lc1dOU0NvbG9yohIUWE5TT2JqZWN0XxAPTlNLZXllZEFyY2hp
dmVy0RcYVHJvb3SAAQgRGiMtMjc7QUhOW2KMjpCVoKmxtL3P0tcAAAAAAAABAQAAAAAA
AAAZAAAAAAAAAAAAAAAAAAA2Q==
**</data>**
**<key>ANSIBrightBlueColor</key>**
**<data>**
**YnBsaXN0MDDUAQIDBAUGFRZYJHZlcnNpb25YJG9iamVjdHNZJGFyY2hpdmVyVCR0b3AS**
**AAGGoKMHCA9VJG51bGzTCQoLDA0OVU5TUkdCXE5TQ29sb3JTcGFjZVYkY2xhc3NPEBww**
**LjQwNzg0MzEzNzMgMC44MzUyOTQxMTc2IDEAEAEAKAAtIQERITWiRjbGFzc25hbWVYJGNs**
**YXNzZXNNXTlNDb2xvcqISFFhOU09iamVjdF8QD05TS2V5ZWRBcmNoaXZlctEXGFRyb290**
**gAEIERojLTI3O0FITltigYOFipWepqmyxMfMAAAAAAAAQEAAAAAAAAAGQAAAAAAAAA**
**AAAAAAAAM4=**
**</data>**
<key>ANSIBrightCyanColor</key>
<data>
YnBsaXN0MDDUAQIDBAUGFRZYJHZlcnNpb25YJG9iamVjdHNZJGFyY2hpdmVyVCR0b3AS
AAGGoKMHCA9VJG51bGzTCQoLDA0OVU5TUkdCXE5TQ29sb3JTcGFjZVYkY2xhc3NPEBww
Ljc4MDM5MjE1NjkgMSAwLjk5MjE1Njg2MjcAEAKAAtIQERITWiRjbGFzc25hbWVYJGNs
YXNzZXNNXTlNDb2xvcqISFFhOU09iamVjdF8QD05TS2V5ZWRBcmNoaXZlctEXGFRyb290
gAEIERojLTI3O0FITltigYOFipWepqmyxMfMAAAAAAAAQEAAAAAAAAAGQAAAAAAAAA
AAAAAAAAM4=
</data>
<key>ANSIBrightGreenColor</key></div> |
| Code | <div>declare(strict_types=1);

namespace PhpSlang\Match\When;

class TypeOf extends AbstractWhen
{
    /**
     * @param $subject
     *
     * @return bool
     */
    public function matches($subject): bool
    {
        return $subject instanceof $this->case;
    }
}</div> |
| Logs | <div>AF_22VFbhaqGevttreUjFreivprRRR,%object
b__MGPA7naqebvq23OaFbhaqGevttreUjFreivprR0_AF_11OaVagresnprVAF_22VFbhaq
GevttreUjFreivprRRR:
.space __SIZEOF_POINTER__
.text
.globl b__MA7naqebvq12FbhaqGevttre12thvqGbFgevatRCX12nhqvb_hhvq_fCpz
.type b__MA7naqebvq12FbhaqGevttre12thvqGbFgevatRCX12nhqvb_hhvq_fCpz,
%function
b__MA7naqebvq12FbhaqGevttre12thvqGbFgevatRCX12nhqvb_hhvq_fCpz:
nop
.data
.globl b__MGIA7naqebvq14OcFbhaqGevttreR
.type b__MGIA7naqebvq14OcFbhaqGevttreR,%object
b__MGIA7naqebvq14OcFbhaqGevttreR:
.space __SIZEOF_POINTER__
.data</div> |

*Table 8.* **Representative High-Influence Training Samples.** We showcase three actual top-ranked examples mined from the training corpus. They exhibit distinct long-range repetitive structures (highlighted in bold) across different modalities.

| Type | Sample Content (Truncated) |
|---|---|
| **LaTeX** | |

```
Q:

Simple example of product not preserving coequaliser in $\mathbf{Top}$

In the category of topological spaces ($\mathbf{Top}$), products do not
always preserve colimits. If they did then $\mathrm{Hom}_\mathbf{Top}
(-\times X,S)$would be representable and hence $\mathbf{Top}$ would be
Cartesian closed(which itisn't). I think that products do preserve
coproducts, so it must be thatthere's some coequaliser which products
don't preserve. I'm trying to understand why this is in more concrete
terms, but I've struggled to find a simple example that I can examine in
detail. What are some simple spaces $A$, $B$ and $X$ and maps $f,g:A\to
B$ in $\mathbf{Top}$ such that the product of $X$ with the coequaliser is
different from the coequaliser of the products?

The same question for the category of locales is here.

A:

(Adapted from Ronald Brown's 'Topology and Groupoids', Section 4.3
Example 4, Page 111.)
Consider $\mathbb{Z}$, $\mathbb{Q}$ and $\mathbb{R}$ with their usual
topologies. Let $i:\mathbb{Z}\hookrightarrow\mathbb{R}$ be the usual
inclusion, and define $j :\mathbb{Z}\to\mathbb{R}$ by $j(n) = i(n+1)$.
Our example of failed preservation will be that the canonical map
$$\mathrm{coeq}(i\times\mathbb{Q},j\times\mathbb{Q})\to\mathrm{coeq}
(i,j)\times\mathbb{Q}$$
is not a homeomorphism.
```

**Database**

**["44.2094250","28.6460842","Law","Bachelor","Romanian","Spiru Haret University","?we=module.fp.searchStudyŎ026ddpN=10904236920026igfm=goto Overview Ŏ026iemuProgramStudiuId=14420026wtok=Ŏ026wtkps=hZDRDoIwDEXZe8 oa7d1q 9gTPwCDBI0IkQ0izH+u2BkOBcY29PbptTsFT86BlZ9IdSrHpWOQuoqmvjdxacIz hewNfe+ 5OpdSbvRSuvu+xSFNjTyEvDoiy782a71AiKjAEYc8cimoc7I7dOUBTaQDmdW0N k34vxrTjRn8IkOHciOYcWdU4T+pNMnUnwq1OSdAa1mtE4CZ0pcDYJiNKCJPo4iuZg8j1ZZ LIaLKALgiKk2AyAQILv59eaNrydtovqm5Rt82QPl8=Ŏ026wchk=e4dd399345f259e2ca15b 9b9ddd178e5e24c86ca", "Law","Faculty of Legal and Economic Sciences"],**
["44.2094250","28.6460842","Law","Bachelor","Romanian","Spiru Haret University","?we=module.fp.searchStudyŎ026ddpN=10904236920026igfm=goto Overview Ŏ026iemuProgramStudiuId=14420026wtok=Ŏ026wtkps=hZDRDoIwDEXZe8 oa7d1q 9gTPwCDBI0IkQ0izH+u2BkOBcY29PbptTsFT86BlZ9IdSrHpWOQuoqmvjdxacIz hewNfe+ 5OpdSbvRSuvu+xSFNjTyEvDoiy782a71AiKjAEYc8cimoc7I7dOUBTaQDmdW0N k34vxrTjRn8IkOHciOYcWdU4T+pNMnUnwq1OSdAa1mtE4CZ0pcDYJiNKCJPo4iuZg8j1ZZ LIaLKALgiKk2AyAQILv59eaNrydtovqm5Rt82QPl8=Ŏ026wchk=e4dd399345f259e2ca15b 9b9ddd178e5e24c86ca", "Law","Faculty of Legal and Economic Sciences"],

**Meaningless**

**Aczg**AczgAczgAczgAczgAczgAczgAczgAczgAczgAczgAczgAczgAczgA
czgAczgAczgAczgAczgAczgAczgAczgAczgAczgAczgAczgAczgAczgAczgA
czgAczgAczgAczgAczgAczgAczgAczgAczgAczgAczgAczgAczgAczgAczgA
czgAczgAczgAczgAczgAczgAczgAczgAczgAczgAczgAczgAczgAczgAczgA
czgAczgAczgAczgAczgAczgAczgAczgAczgAczgAczgAczgAczgAczgAczgA
czgAczgAczgAczgAczgAczgAczgAczgAczgAczgAczgAczgAczgAczgAczgA
czgAczgAczgAczgAczg**AAAAAAA**AAAAAAAAAAAAAAAAAAAAAAAAAA
AAAAAAAAAAAAAAAAAAAAAAAAAAAAAAAAAAAAAAAAAAAAAAAAAAAAAAAAAA
AAAAAAAAAAAAAAAAAAAAAAAAAAAAAAAAAAAAAAAAAAAAAAAAAAAAAAAAAA
AAAAAAAAAAAAAAAAAAAAAAAAAAAAAAAAAAAAAAAAAAAAAAAAAAAAAAAAAA
AAAAAAAAAAAAAAAAAAAAAAAAAAAAAAAAAAAAAAAAAAAAAAAAAAAAAAAAAA
AAAAAAAAAAAAAAAAAAAAAAAAAAAAAAAAAAAAAAAAAAAAAAAAAAAAAAAAAA
AAAAAAAAAAAAAAAAAAAAAAAAAAAAAAAAAAAAAAAAAAAAAAAAAAAAAAAAAA
AAAAAAAAAAAAAAAAAAAAAAAAAAAAAAAAAAAAAAAAAAAAAAAAAAAAAAAAAA
AAAAAAAAAAAAAAAAAAAAAAAAAAAAAAAAAAAAAAAAAAAAAAAAAAAAAAAAAA
AAABABgAAAAAAAAMAAAAAAAAAAAAAAAAAAAAAAAA**Aczg**AczgAczgAc

