# OpenReview forum: "Mechanistic Data Attribution: Tracing the Training Origins of Interpretable LLM Units"
_ICML.cc/2026/Conference — ICML 2026 spotlight_

### Official Review · Reviewer_opn9 · 2026-03-06

**Soundness:** 3
**Presentation:** 3
**Significance:** 3
**Originality:** 2
**Overall Recommendation:** 5
**Confidence:** 3

**Summary:**

This paper applies a data attribution method (EK-FAC) to determine the pretraining data responsible for the emergence of induction heads; they do this across various small Pythia models. They find that repetitive code / LaTeX / XML data causes their emergence, and that heads largely agree about which data was most important to their emergence. The authors show that removing this sort of data stalls the heads emergence; via a data augmentation pipeline, they also show that adding more of this sort of data can spur the heads' emergence.

**Compliance With Llm Reviewing Policy:**

Affirmed.

**Final Justification:**

I'm satisfied by the rebuttal, and have positively updated my assessment.

**Key Questions For Authors:**

- How does this method scale? Is it possible to scale this up to larger models with more interesting mechanisms?
- What are the requirements for a mechanism to admit attribution via this method?
- Do you have any insights into how interventions like the ones your performed might affect the long-term training trajectory / abilities of models?

**Limitations:**

yes

**Strengths And Weaknesses:**

**Soundness**: This paper's experiments, especially the causal ones, are relatively convincing. I think that actually retraining the models lends a lot of credence to the paper's claims, although I'm a bit surprised to see the small number of steps on which the authors trained their models. I also think that the authors could more effectively argue for their technique's effectiveness if they used it to investigate at least one other head, e.g. successor heads.

One issue that goes underdiscussed is the scalability of this technique. The authors only work with small models <= 160m, which I assume is because this method does not scale (otherwise, I'd love to see results on OLMo). How bad is this method's scaling? Is there a path towards applying it to larger models? It is my impression that this is one of the key problems with this and other data attribution methods.

**Presentation**: This paper's presentation is fine, with some minor writing issues, though:
- Only proper nouns should be capitalized, even if they form part of an acronym. That means "large language models" and "influence functions" are correct.
- Though induction is rather simple, this paper's definition thereof (lines 159-) is rather complicated. It can be simplified.

**Significance**: I think that this work has the potential to be significant. I know people who are interested in the emergence of specific mechanisms, and would love to have a method that does what this method does. However, I think that there are a couple of bottlenecks to practical impact. In particular, precious few global mechanisms (like induction heads) are known to exist in models. Moreover, the authors don't explore whether the faster emergence of induction heads caused by their interventions has a meaningful impact on downstream performance (an issue explored in prior work, which found that accelerating specific heads' development actually hurt performance.).

**Originality**: This analysis is relatively novel: I don't know of any work using data attribution methods to find the source of specific mechanisms, and I think it's an interesting question. However, this paper either fails to cite / discuss or inadequately discusses work on interpretability over time and induction heads. Re: the first point, the paper doesn't cite any work by Naomi Saphra, who has worked quite a bit on this topic. I think the authors would do well to read all of her work, but [Chen et al. (2024)](https://arxiv.org/abs/2309.07311) is particularly relevant: the authors trace the emergence of specific syntax heads over time, and perform ablations where they manipulate the data on which the model is trained, boosting and impairing the formation of these heads. Crucially, sometimes this helps the model's overall performance, sometimes not.

Beyond that body of work, I feel this paper spends a bit too little time on existing interp over time work. For example Tigges et al study more than just induction (and this paper would do well to do so too!). And while the other induction over time papers mentioned mostly study toy settings, this paper would benefit from discussing how their findings match up with prior work. While finding the specific documents that cause induction circuit formation is valuable, some of the other things that the authors claim to find (that PT heads are necessary for induction circuits, or indeed that repetitive data (as found in code and LaTeX) causes incution heads' formation) are less novel. I'd also suggest discussing Crosbie et al. (2024).

Alternatively, the authors could have studied a mechanism that is much less well-known; this would have been a nice way to highlight the ways in which this method can explain mechanisms that were previously poorly understood.

Overall, I think that this paper is somewhat flawed, but that the technique + problem combination seems interesting enough. I wouldn't mind this paper being accepted, so long as the authors improve their engagement with existing literature, and make the limitations of this work clearer. That said, I don't feel very strongly.

---

> ### Author Rebuttal · Authors · 2026-03-31
>
> We especially appreciate your recognition that the retraining-based experiments provide relatively convincing causal evidence, as well as your view that the core technique-problem combination is novel and potentially significant.Below we address each point in turn.
>
> ## W1. Short training horizon
> Please refer to the reply regarding reviewer iYq2's W1
>
> ## W2&W7. Limited evaluation beyond induction / previous-token heads, lack of other mechanisms
> We selected induction heads as a canonical case study because the primary goal of MDA is not to discover new mechanisms post-hoc, but to provide a complementary framework for studying well-established mechanisms from a training-data perspective. As you noted, there is significant interest in the emergence of specific mechanisms; we believe the utility of MDA will continue to expand alongside the broader field of Mechanistic Interpretability (MI) and the discovery of new mechanism. While we agree that applying MDA to more complex mechanisms is a compelling direction, we intend to explore this in future work. To further demonstrate the generalizability of MDA, we add experiments on **SAE features** and find meaningful results, for the results please see https://anonymous.4open.science/r/ICML-rebuttal-code-0B5C/OLMo%20Influence%20Samples.pdf.
>
> ## W3&Q1. Scalability to larger models
> In response to the concern regarding scalability, we have added empirical evidence using **OLMo-2-1B and 7B** (https://anonymous.4open.science/r/ICML-rebuttal-code-0B5C/OLMo%20Influence%20Samples.pdf). The runtime statistics confirm an almost linear relationship between computational cost and parameter count. For more details, please refer to the reply regarding reviewer KKRk's W1&Q1.
>
> ## W4. Presentation issues
> Thank you for these helpful comments. We will revise the writing accordingly.
>
> ## W5&Q3. Downstream performance impact
> This is an important concern, and we agree that faster emergence of a mechanism does not automatically imply better overall model performance. To offer preliminary insights into this aspect, we have incorporated evaluations of **language modeling loss** and **factual knowledge acquisition** within our specific investigation window, in which we don’t find significant negative side effects (https://anonymous.4open.science/r/ICML-rebuttal-code-0B5C/eval_loss_curves.pdf).
>
> ## W6. Engagement with related literature
> Thank you for this point. We agree that the paper should engage more thoroughly with prior work on interpretability over time and induction-head emergence. We will improve the related-work discussion in the revision.
> We would also like to clarify that we do not “claim to find PT heads are necessary for induction circuits”. Rather, we take this as known and deliberately design our setup to disentangle their interaction. Specifically, we intentionally offset the timing—the point at which we identify the previous token head is chosen to be just before the emergence of induction—precisely to separate their interaction. We also employ different probing strategies for this purpose(details in Table 3 and 4). Regarding the claim that the emergence of induction heads is driven by repeated data, we are indeed aware that prior work has demonstrated this in controlled settings. However, our claim is that MDA provides a way to identify, within **real pretraining data**, which individual samples most influenced the emergence of a specific interpretable mechanism, and that these attributions support targeted causal interventions. In this sense, our contribution is less about discovering that repetition matters in the abstract, and more about tracing mechanism formation back to concrete training examples in realistic data. Relatedly, we have added stronger control experiments showing that even after matching for simple repetition statistics such as bigram and trigram counts, MDA-based selection continues to outperform these controls. For these results, please refer to our rebuttal to Reviewer AKvd’s Q3. This suggests that the signal captured by MDA is not reducible to simple surface repeat metrics alone.
>
> ## Q2.Type of mechanism suitable for our method
> Mechanistically, our approach is most suitable when the target behavior is primarily realized by a relatively localized subset of parameters. In that setting, restricting attribution to the corresponding subspace reduces both computational cost and noise from unrelated parameters, while still capturing most of the mechanism-relevant signal. Mathematically, that equals to constructing a probing function such that the dot product between its gradients and Hessians with respect to the targeted parameters is orthogonal(or in a lower order).(For details, please refer to our rebuttal to Reviewer iYq2’s Q3)

---

> > ### Author Rebuttal · Reviewer_opn9 · 2026-04-01
> >
> > Thanks! I think this rebuttal has resolved most of my concerns. I think that the improved discussion of related work and additional experiments will strengthen this paper quite a bit.

---

> > > ### Author Response · Authors · 2026-04-01
> > >
> > > Thank you for your support and the positive feedback. We appreciate your insightful suggestions throughout the review process, which have improved the quality and clarity of our paper.

---

### Official Review · Reviewer_KKRk · 2026-03-09

**Soundness:** 4
**Presentation:** 3
**Significance:** 3
**Originality:** 3
**Overall Recommendation:** 5
**Confidence:** 3

**Summary:**

A central theme addressed by this manuscript is understanding where interpretable components inside large language models actually come from during the training process. The manuscript aims to address an important concept by linking specific training data samples to the emergence of induction heads and previous token heads. The author proposes a framework called Mechanistic Data Attribution which uses influence functions to find out which training examples cause these heads to form. The author tests this on the Pythia model family and shows that highly repetitive data acts as a catalyst for induction heads. Then the author builds a data augmentation pipeline to generate synthetic data that speeds up the formation of these circuits.

**Compliance With Llm Reviewing Policy:**

Affirmed.

**Final Justification:**

This paper introduces a neat idea — instead of asking which data caused this output, it asks which data caused this mechanism to form. The causal validation through data deletion and augmentation is strong, and the finding that repetitive structured data catalyzes induction heads is both intuitive and useful.

The rebuttal fully addressed my concerns. The main limitation is that the strongest causal evidence comes from smaller models, and it remains to be seen how well this generalizes to much larger models with more complex circuits. The augmentation pipeline relying on an external LLM also adds some opacity. But the core contribution is solid and fills an important gap.

**Key Questions For Authors:**

Q1: How well do you think this Mechanistic Data Attribution framework scales to models with over seven billion parameters where compute costs for influence functions blow up?
Q2: Could the author explain if there are any risks of negative side effects when injecting this synthetic repetitive data like maybe it hurts other capabilities while helping induction heads?
Q3: Did the author notice any differences in how the models learned factual knowledge when the induction heads formed earlier due to the synthetic data?

**Limitations:**

yes

**Strengths And Weaknesses:**

Soundness
Strength:
The experimental design using data deletion and augmentation is very solid. It provides causal evidence rather than just correlations. The math behind the approximation seems correct and well adapted for joint subspaces like query and key matrices.
Weakness:
The evaluation only looks at smaller models up to 160M parameters. It is not entirely clear if these exact findings scale up to multibillion parameter models where circuit behavior gets much more complex.

Presentation
Strength:
The paper is really easy to read and flows well. The figures especially the ones showing the phase transitions of induction scores are clear and help explain the text perfectly.
Weakness:
Some of the mathematical derivations in the appendix could be explained a bit more simply in the main text. Sometimes the author jumps into heavy math without enough intuitive setup.

Significance
Strength:
This work bridges a huge gap between mechanistic interpretability and training data attribution. Giving developers a way to steer model behavior by curating data for specific circuits is a big deal for AI safety and efficiency.
Weakness:
The practical data augmentation pipeline relies on an external language model to extract patterns which adds a layer of opacity. If the extraction misses subtle patterns the synthetic data might not capture everything important.

Originality
Strength:
Applying influence functions directly to internal model components rather than the final output loss is a very clever and novel idea. The whole pipeline from attribution to automated synthetic data generation feels fresh and highly original.

---

> ### Author Rebuttal · Authors · 2026-03-31
>
> We appreciate your recognition of the paper’s causal validation through deletion and augmentation experiments, your positive assessment of the mathematical treatment for joint subspaces, and your encouraging comments on the paper’s clarity, significance, and originality. Below we address your questions and concerns in turn.
>
> ## W1&Q1. Scaling beyond small models
> The main retraining-based validation experiments are conducted on models up to 160M parameters. This choice is primarily due to computational cost: our strongest causal evaluations require **repeated retraining and controlled comparisons**, which are expensive for academic community. However, MDA relies on scalable influence-function approximations (EK-FAC, SOTA in the domain), and because our attribution is restricted to mechanism-specific parameter subspaces, we can edit the computation graph so that it can be substantially more efficient than full-model tracing.
> In response to the concern regarding scalability, we have added empirical evidence using **OLMo-2-1B and 7B** (https://anonymous.4open.science/r/ICML-rebuttal-code-0B5C/OLMo%20Influence%20Samples.pdf). The runtime statistics confirm an almost **linear relationship between computational cost and parameter count**. Critically, because the tracing process is fully data-parallelizable, the wall-clock time scales inversely with the number of available GPUs, ensuring that our framework can be efficiently deployed on high-performance computing clusters.
> | Model | Layers | Model Dim | Heads |GPU hours(A100)|
> |-------|------|---------|-----|---|
> |     Pythia-14M  |    6    |     128      |    4   |  11.6 |
> |    Pythia-31M    |    6    |      256     |    8   | 14  |
> |      Pythia-70M  |    6    |      512     |   8    |  16.8 |
> |     Pythia-160M  |    12    |      768     |   12    |  47.3 |
> |    OLMo-2-1B    |    16    |      2048     |  16     | 80  |
> |    OLMo-2-7B  |    32    |      4096     |    32   | 384  |
>
> ## W2. Heavy mathematical presentation
> Thank you for this helpful suggestion. We agree that some of the derivations in the appendix would benefit from more intuitive setup in the main text. In the revision, we will improve the exposition by adding more intuition before the formal derivations and by simplifying the presentation of the key mathematical steps where possible.
>
> ## W3. Opacity of the external LLM in the augmentation pipeline
> In our current experiments, the pipeline is effective enough to produce measurable gains, but we agree that it is not yet fully transparent or guaranteed to capture all subtle mechanism-relevant patterns. We also see this as an important direction for future work, including more controlled extraction procedures, improved diversity constraints, and potentially human verification or other interpretability-guided pattern selection methods.
>
> ## Q2 & Q3. Possible negative side effects of synthetic repetitive data
> Given that our interventions are applied during a relatively early and concentrated phase of pre-training, a comprehensive long-term capability assessment falls beyond our current computational scope. To offer preliminary insights into this aspect, we have incorporated evaluations of **language modeling loss** and **factual knowledge acquisition** within our specific investigation window, in which we don’t find significant negative side effects (https://anonymous.4open.science/r/ICML-rebuttal-code-0B5C/eval_loss_curves.pdf).

---

> > ### Author Rebuttal · Reviewer_KKRk · 2026-04-01
> >
> > The authors have addressed my concerns. I will raise my score accordingly. Good luck!

---

> > > ### Author Response · Authors · 2026-04-01
> > >
> > > Thank you for your support and for the increased score. Your feedback has been very helpful in improving our work. We appreciate the well wishes!

---

### Official Review · Reviewer_AKvd · 2026-03-09

**Soundness:** 3
**Presentation:** 3
**Significance:** 3
**Originality:** 3
**Overall Recommendation:** 5
**Confidence:** 4

**Summary:**

This paper introduces Mechanistic Data Attribution (MDA), a framework that attributes the interpretable internal units (e.g., induction heads) to specific training samples. MDA adapts the influence function to a unit specific probe objective and uses scalable curvature approximations (EK-FAC) to efficiently estimate per-sample influence. The authors then causally validate the attribution by retraining after deleting or augmenting high influence samples, showing targeted interventions modulate the formation of the corresponding heads, while random interventions do not. They further analyze influential samples (often repetitive, structured text such as LaTeX/XML), show that influence is heavy-tailed, and argue that shifting induction-head formation also shifts in-context learning (ICL) metrics, providing causal support for the induction head/ICL link. Finally, they propose a “mechanistic data augmentation” pipeline that extracts patterns from influential samples and synthesizes more such data to accelerate circuit formation across scales.

**Compliance With Llm Reviewing Policy:**

Affirmed.

**Final Justification:**

The authors have adressed my concerns by rebuttal, i think it should be accepted.

**Key Questions For Authors:**

Questions:
Here are some questions I am concerned about:

1. Ranking stability: How stable are top influential samples across seeds, checkpoints, and modest probe variations? Can you report rank correlations/overlap curves?
2. Any preliminary results for MLP neurons or SAE features?
3. If you construct a control set matched on length/domain/entropy and bigram-repeat statistics, does targeted augmentation still outperform?

**Limitations:**

The idea is novel and well-motivated, and the causal retraining interventions are unusually compelling for attribution work. You may improve your paper from:

1. Add a robustness section: probe variants, different unit definitions, and approximation quality vs intervention success.
2.  Add stronger matched controls for deletion/augmentation to rule out “domain shift” explanations.

After these improvements, I think this will be a strong paper for publication

**Strengths And Weaknesses:**

Strengths:
1. Clear conceptual contribution: The paper moves from which data causes this output to which data causes this mechanism, bridging mechanistic interpretability and data attribution in a clean way.
2. Interesting Finding: The finding that repetitive structured “noisy” data (e.g., LaTeX/XML-like patterns) acts as a catalyst is both intuitive and operationally useful for dataset design.
3. Practical pipeline: The proposed pattern extraction + synthesis to steer training dynamics is a compelling application direction beyond analysis.
4. Causality Validation: The paper intervenes in pretraining data and shows predictable effects on head emergence.

Weakness:
1. Narrow mechanistic scope in validation: The strongest causal evidence is for induction and previous-token heads on Pythia-family runs. It’s not yet clear how well MDA generalizes to more distributed or less localized mechanisms
2. Probe-design sensitivity: The results depend heavily on the chosen probe metric (Table 3); I think you should add some robustness test for the selected probe function/metric.

---

> ### Author Rebuttal · Authors · 2026-03-31
>
> We appreciate your recognition of the paper’s conceptual contribution, the practical value of the mechanistic data augmentation pipeline, and the strength of the causal validation through retraining interventions. We also thank you for the constructive suggestions on robustness, broader mechanism coverage, and stronger matched controls. Below we address each point in turn.
>
> ## W1&Q2. Narrow mechanistic scope in validation
> We have added supplementary experiments on a **different model family** (OLMo-2-1B and OLMo-2-7B’s induction heads) and on **additional unit types** (SAE features of Pythia 70M). These new results, reported in https://anonymous.4open.science/r/ICML-rebuttal-code-0B5C/pythia-70m%20SAE.pdf and https://anonymous.4open.science/r/ICML-rebuttal-code-0B5C/OLMo%20Influence%20Samples.pdf, providing evidence that MDA extends beyond Pythia attention heads and remains effective across both model architectures and mechanism definitions.
>
> ## W2&Q1. Probe-design sensitivity
> We have conducted a robustness analysis covering various aspects of probe design, including different **probing functions** (e.g., logits-based vs. prefix-matching attention-based probes) and **probing datasets** (constructed using different random seeds). Our results show high consistency across datasets, with both **Top-k overlap** and **Spearman rank correlation** reaching approximately 0.9. However, the prefix-matching attention-based probe exhibited significant discrepancies. We also performed interventions (augmentation) on the samples identified by this probe, and the results show that it is even weaker than the baseline. A plausible explanation is that induction heads cannot be fully characterized by their attention patterns alone; the copying functionality mediated by the OV matrix is also indispensable. These findings suggest that the logits-based probe is a more robust and comprehensive design choice. For full results, see https://anonymous.4open.science/r/ICML-rebuttal-code-0B5C/rank%20robustness.pdf and https://anonymous.4open.science/r/ICML-rebuttal-code-0B5C/controlled%20baseline%20&%20probe%20variant.pdf.
>
> ## Q3. Matched controls for targeted augmentation
> This is a well-founded concern, and we agree that stronger controls are important for ruling out alternative explanations such as domain shift or trivial distributional differences.
> In response, we constructed better-matched control sets by accounting for properties including **sample length, entropy, domain, and n-gram statistics (including bigram and trigram repetition)**. We then used these matched controls as additional baselines in the augmentation experiments; the corresponding results are provided in https://anonymous.4open.science/r/ICML-rebuttal-code-0B5C/controlled%20baseline%20&%20probe%20variant.pdf. These new baselines strengthen the conclusion that the gains from targeted augmentation are not solely explained by “domain shift”, but instead reflect signal captured by MDA beyond these coarse corpus-level statistics.

---

> > ### Author Rebuttal · Reviewer_AKvd · 2026-04-01
> >
> > Interesting work, I will increase my score to 5; Good luck

---

> > > ### Author Response · Authors · 2026-04-01
> > >
> > > We are glad that we successfully addressed your concerns. We also appreciate your encouraging words and the "Good luck"—they are much appreciated as we continue to refine this work.

---

### Official Review · Reviewer_iYq2 · 2026-03-13

**Soundness:** 2
**Presentation:** 3
**Significance:** 3
**Originality:** 3
**Overall Recommendation:** 5
**Confidence:** 2

**Summary:**

This paper propose Mechanistic Data Attribution, a framework for tracing training data origins of specific interpretable model components (e.g., induction heads) in LLMs using influence functions _restricted_ to mechanism-specific parameter subspaces. The authors validate the method via data deletion and augmentation experiments across Pythia models.

**Compliance With Llm Reviewing Policy:**

Affirmed.

**Final Justification:**

see the rebuttal acknowledgement

**Key Questions For Authors:**

Q1. [`Restricted tracing vs Full tracing] when you restrict the parameter space to mechanism specific area and run the influence function tracing back to training data, how big is the difference between the restricted tracing versus the full tracing? is the difference roughly the same on different mechanisms?

Q2. [Alternative Baseline via Direct Path Extraction] Another potential way to trace mechanisms back to data would be to explicitly isolate the computational path corresponding to a given mechanism (e.g., a specific attention head), and perform input–output analysis on that path directly. For instance, one could derive an n-gram–style description of that isolated path’s behavior and compare it to patterns in the training data. This would effectively turn the mechanism into a textual or symbolic sketch, which could then be matched against corpus statistics.

Have you considered or implemented such a baseline? How does MDA compare to directly extracting and analyzing mechanism-specific input–output behavior in this way?

Q3. [Restriction Approximation] In MDA, the restriction to a parameter subspace (e.g., a specific head) effectively “carves out” the computation into two parts: (i) the mechanism of interest, and (ii) the remaining network.

However, in the full model these parts interact nonlinearly and jointly determine behavior. How do you quantify the interaction between the restricted subspace and the rest of the model during attribution? In particular, does the restriction implicitly approximate cross-component interactions at a lower (e.g., zeroth) order?

If so, how might this affect attribution accuracy, especially for mechanisms that rely on coordinated multi-head or cross-layer dynamics? It would also be helpful to clarify how this approximation compares conceptually to functional decomposition approaches (e.g., Decomposing LLM Computation with Jets, ICLR 2026, ICLR 2026), which explicitly separate polynomial paths and nonlinear remainders.

**Limitations:**

yes

**Strengths And Weaknesses:**

Strengths

- timely topic in terms of connecting mechanistic interpretability to training-time causality.
- sensible method of adapting influence functions to component-level attribution.
- the figures are visually pleasant

Weaknesses

- Influence analysis is limited to predefined training windows.

- Robustness to curvature approximation (EK-FAC) could be further analyzed.

- experiment is mostly on attention heads; more mechanism coverage would strengthen generality.

Related work

This work is conceptually related to recent efforts that treat interpretability as a structural decomposition problem rather than purely data-driven probing (Decomposing LLM Computation with Jets, ICLR 2026). While that line of work focuses on functional expansion of inference-time computation, this paper traces training-time causal origins of specific mechanisms. The two approaches appear complementary, and briefly discussing this relationship would help the paper’s positioning within the emerging “structural interpretability” paradigm.

---

> ### Author Rebuttal · Authors · 2026-03-31
>
> Thank you for your recognition of our work and your valuable suggestions. Below we address your concerns and questions in turn.
>
> ## W1. Limited to predefined training windows
> Conceptually, our framework itself is not limited to specific training windows. However, given computational resource constraints, we carefully selected the window in which the mechanisms we study (especially induction heads) are **fully formed and identifiable (see Appendix Figure 6)**. We agree that exploring longer periods to study the long-term effects of interventions is a worthwhile direction which we recognize as a future research goal.
>
> ## W2. Robustness to the EK-FAC curvature approximation
> We agree that the robustness of the curvature approximation is important. In our experiments, EK-FAC is used because it is currently the SOTA approximations for influence-function-style analyses at LLM scale. To assess robustness, we repeated the fitting procedure with different random seeds and different fitting datasets, and found that the resulting approximations were highly consistent, with **average cosine similarity around 0.98 between the corresponding matrices**.
>
> ## W3. Limited mechanism coverage beyond attention heads
> Thank you for your suggestion. We have conducted supplementary experiments on **SAE features** from Pythia 70m and obtained meaningful results, proving that our method generalizes well. Due to the characters limitation, the results can be seen in https://anonymous.4open.science/r/ICML-rebuttal-code-0B5C/pythia-70m%20SAE.pdf
>
> ## Q1. Restricted tracing vs. full tracing
> Our work focuses on Mechanistic Interpretability (MI) rather than Training Data Attribution (TDA). While our goal is to investigate the **formation mechanisms of specific interpretable units**, TDA methods like "full tracing" are designed to map the influence of training data onto model behavior. These two paradigms are fundamentally complementary rather than mutually exclusive; therefore, a direct performance comparison is not applicable as they address distinct research questions.
>
> ## Q2. Alternative baseline via direct path extraction
> The concept of isolating a mechanism-specific computational path and characterizing it through input-output behavior offers a complementary perspective to MDA. Our current approach aims to trace the causal influence of training on parameters associated with a specific mechanism, supported by mathematical guarantees. It would indeed be insightful to investigate whether the baseline method exhibits similar causal influence; we will incorporate this comparison into the revised version. We have also added a new baseline, primarily motivated by Reviewer Akvd’s request (Q3), which also partially overlaps with your suggestion(on n-gram repetition). Please refer to that section for details.
>
> ## Q3. Restriction approximation and cross-component interactions
> Currently, MDA does not offer a pre-emptive method for estimating the relevant interaction order, which we acknowledge as a pivotal direction for future research. To address this in the present study, we have included experiments to evaluate the contribution of parameters outside the restricted subspace (https://anonymous.4open.science/r/ICML-rebuttal-code-0B5C/cross_head_influence.pdf). These results quantify the fidelity of our approximation; specifically, they demonstrate that the influence samples of induction heads have a minimal impact on non-target components, confirming that our approach successfully captures the core mechanism.
> We also appreciate the reference to *Decomposing LLM Computation with Jets*. We find that work highly valuable and complementary to ours. While Jets provides a principled framework for characterizing functional structures from an observational perspective, it could potentially assist in estimating interaction orders or predicting the reliability of restriction-based approximations. Integrating these insights remains a key goal for our future work.

---

> > ### Author Rebuttal · Reviewer_iYq2 · 2026-04-04
> >
> > Thank you for the clear and thoughtful rebuttal. The additional experiments (EK-FAC robustness, cross-component influence, and SAE features) address my main concerns and improve my confidence in the method.
> >
> > I also appreciate the discussion of alternative baselines and the connection to related work (e.g., Jets). I encourage the authors to incorporate these clarifications and literature positioning into the final version.
> >
> > Based on the rebuttal and the anticipated revisions, I am updating my score from **4 to 5**.

---

> > > ### Author Response · Authors · 2026-04-04
> > >
> > > Thank you for your positive feedback! We are happy to address your concerns.

---

### Decision · Program_Chairs · 2026-04-30

**Decision:**

Accept (spotlight)

**Comment:**

This paper attributes specific internal mechanisms within a model to specific elements of the training data and tests their claims with rigorous counterfactual training scenarios. Given the increasing number of people interested in studying circuit formation, this is a timely objective. The approach is original and the validation is satisfactory.

Most weaknesses discussed are minor presentation issues, with some reviewers requesting more discussion of related works. These are all addressable by the camera ready deadline.